# A PSA SNP associates with cellular function and clinical outcome in men with prostate cancer

Genetic variation at the 19q13.3 *KLK* locus is linked with prostate cancer susceptibility in men. The non-synonymous *KLK3* single nucleotide polymorphism (SNP), rs17632542 (c.536 T > C; Ile163Thr-substitution in PSA) is associated with reduced prostate cancer risk, however, the functional relevance is unknown. Here, we identify that the SNP variant-induced change in PSA biochemical activity mediates prostate cancer pathogenesis. The 'Thr' PSA variant leads to small subcutaneous tumours, supporting reduced prostate cancer risk. However, 'Thr' PSA also displays higher metastatic potential with pronounced osteolytic activity in an experimental metastasis in-vivo model. Biochemical characterisation of this PSA variant demonstrates markedly reduced proteolytic activity that correlates with differences in in-vivo tumour burden. The SNP is associated with increased risk for aggressive disease and prostate cancer-specific mortality in three independent cohorts, highlighting its critical function in mediating metastasis. Carriers of this SNP allele have reduced serum total PSA and a higher free/total PSA ratio that could contribute to late biopsy decisions and delay in diagnosis. Our results provide a molecular explanation for the prominent 19q13.3 *KLK* locus, rs17632542 SNP, association with a spectrum of prostate cancer clinical outcomes.

Prostate cancer (PCa) is the second most common malignancy in men world-wide. Serum prostate-specific antigen (PSA) has been the common method of PCa diagnosis for decades[1]. Recent randomised trials[2,3] and Screening Trials[4] showed that PSA testing results in reduced PCa-mortality but also leads to over-diagnosis emphasising the need to revise PSA-based screening for PCa to an individualised, risk stratified and informed decision-making model for men, especially at a younger age. PCa diagnosis by the Free/Total (f/t) PSA ratio, which is lower in PCa compared to those with benign prostatic hyperplasia[5–7] and other nomograms such as the 4Kscore[8], are questioned for their clinical utility in discriminating indolent and aggressive PCa and the net benefit these tests add for clinical decision-making[9].

PSA liquefies semen by cleaving semenogelin proteins[10] and has a role in tumour progression by cleaving growth factors, and extracellular matrix (ECM) proteins, increasing migration of PCa cells[11], bone metastasis[12–14] and angiogenesis[15]. Genome-wide association studies (GWAS) to date have confirmed that there are now more than 450 single nucleotide polymorphisms (SNPs) that cumulatively explain 42.6% of the familial component of PCa risk in European ancestry[16–18]. Given the clinical importance of PSA in PCa, we and others have earlier performed fine-mapping at the 19q13.3 locus near the kallikrein related peptidase-3 (*KLK3*) gene encoding PSA and have shown rs17632542, a non-synonymous SNP (amino acid change Ile to Thr at position 163), is the putative causal SNP at this locus associated with reduced PCa risk[19–22]; however, the exact role of PSA in PCa pathogenesis has not been fully elucidated. Genetic factors may contribute to the differences in serum PSA concentrations and genetic correction to PSA levels may lower the frequency of prostate biopsies[11,23–25]. Thus, there has been a conundrum as to whether this association of rs17632542 SNP with PCa risk is due to a true biological role of the SNP in PCa pathogenesis or simply reflects the impact of this SNP on PSA measurement, as cases and disease-free controls recruited in most of the

✉e-mail: jyotsna.batra@qut.edu.au

GWAS studies have a selection bias based on PSA testing being used to detect the disease.

Here, we show that the rs17632542 SNP affects PSA-driven function as seen in in-vitro assays and in-vivo preclinical xenograft models of tumour growth and metastasis. This suggests there is a plausible biological role for the rs17632542 SNP underlying the risk association finding. Using a suite of biochemical assays, we comprehensively show that the SNP leads to an alteration in the proteolytic activity of PSA, which in turn affects the function of PSA in the tumour microenvironment. Our data also indicate that this SNP PSA variant is likely differently detected by the clinically used PSA immunoassays, also affecting the free/total PSA ratio. Furthermore, we explored the association of the rs17632542 variant with PCa risk in three large independent cohorts and identified the SNP to be associated with both PCa risk and survival, but paradoxically, in opposite directions.

## Results

### Thr[163] PSA has reduced effect on PCa cell proliferation and migration

In terms of risk association for rs17632542, the evidence for how this SNP confers risk is still unclear. We thus explored the impact of PSA variants in controlled in-vitro assays. Accordingly, lentivirus vector-based overexpression of furin-activable PSA isoforms of wild type (Wt) PSA, Thr[163] variant (encoded by the rs17632542 SNP [C] allele) and Ala[195] catalytic inactive mutant control (which is an additional control to confirm that the proteolytic activity is important for PSA function); and plasmid vector control (Supplementary Fig. 1A) was performed in androgen receptor- (AR) and PSA-deficient PC-3 and AR- and PSA-expressing LNCaP cell lines (Fig. 1A, Supplementary Fig. 1B). In LNCaP cells, we first conducted lentivirus-mediated short hairpin RNA (shRNA) against *KLK3* (Fig. 1A, Supplementary Fig. 1B) and then re-transfected with the PSA isoforms. For comprehensive validation in patient-derived organoids, we generated an additional cell line model for lentivirus vector-based overexpression of furin-activable PSA isoforms (Wt, Thr[163], Ala[195] and eGFP) (Fig. 1A, Supplementary Fig. 1A) in the AR- and PSA-low MSK3 cell line[26] (Fig. 1A, Supplementary Fig. 1C). PSA (mRNA and protein) levels in these overexpression models was similar to the endogenous expression levels in DUCaP PCa cells.

Expression of Wt PSA in the PC-3 and LNCaP cell lines measured by IncuCyte live cell imaging system showed marked increase in the rate of cell proliferation, while that of Thr[163] PSA did not have any effect (despite their similar expression levels; Supplementary Fig. 1A), suggesting a high functional impact of the SNP (Fig. 1B, C, Supplementary Fig. 1D). As expected, inactive mutant and vector control cells did not show any effect (Fig. 1B, C). As IncuCyte live cell imaging analysis was not suitable for MSK3 cell proliferation analysis, we utilised PrestoBlue viability assays. Consistent to our observation in PC-3 and LNCaP cells, MSK3 cells transfected with Wt PSA variant exhibited higher proliferation compared to Thr[163] PSA and vector transfected cells (Fig. 1D, Supplementary Fig. 1D).

We next investigated the ability of PC-3-PSA cells to migrate using wound healing assays using the IncuCyte live cell imaging system. While the overexpression of Wt PSA enhanced migration of PC-3, the Thr[163] PSA overexpression had no effect (Fig. 1E, Supplementary Fig. 1E). For LNCaP- and MSK3-PSA cells, migration was analysed using Boyden chamber assays. Thr[163] PSA transfected LNCaP and MSK3 cells also exhibited reduced migration compared to all three control groups, including Wt PSA expressing cells (Fig. 1F, G, Supplementary Fig. 1E). Overall, Thr[163] PSA transfection exhibited lower cell proliferation, and migration, thus, lacking the activity of Wt PSA.

### Thr[163] PSA leads to small subcutaneous tumours

Having asserted that the rs17632542 SNP affects the bioactivity of PSA in-vitro, we explored the impact of this PSA variant on primary tumour growth in an in-vivo context. NSG Mice were implanted subcutaneously with luciferase transfected PC-3 cells expressing Wt PSA, Thr[163] PSA or eGFP vector control (Fig. 1H). PC-3-Wt PSA cells developed the largest tumours (by volume [Fig. 1I, J] and weight [Fig. 1K]), as observed by day 38, compared to those implanted with PC-3-Thr[163] PSA cells or vector control PC-3 cells. Necrotic areas were observed in all the tumours (Fig. 1L). Serum concentration of total PSA at endpoint was also highest in mice bearing PC-3-Wt PSA tumours (*P* = 0.01) (Fig. 1M). Collectively, as compared to Thr[163] PSA expressing cells, Wt PSA expression was associated with higher tumour burden in this preclinical primary tumour model, which correlated with reduced PCa risk for the rs17632542 SNP.

### Thr[163] PSA increases invasive ability of prostate cancer cells

As three-dimensional (3D) in-vitro cell culture systems recapitulate in-vivo conditions, we generated spheroids to analyse the proliferation and invasive potential of the PC-3, LNCaP and MSK3 patient-derived organoid cells overexpressing furin-activable PSA variants (Fig. 2A). The spheroids' growth (area of 2D projection and number of spheroids) and invasive ability (circularity/compactness) were analysed (Supplementary Fig. 2A, B). In Matrigel, preformed PC-3 and LNCaP cell aggregates formed single stellate spheroids, characterised by migration of cells through the surrounding Matrigel matrix (Fig. 2B–G). Thr[163] PSA expressing PC-3 and LNCaP spheroids showed a higher number of peripheral invading cells (Fig. 2C, F) and less spherical inner cores (Fig. 2D, G), suggestive of a more invasive phenotype compared to Wt PSA expressing spheroids (Fig. 2B–G). MSK3 cells, seeded as a single suspension in Matrigel, formed multiple small, circular spheroids (Fig. 2H). Thr[163] PSA expressing MSK3 spheroids showed a higher growth potential with higher spheroid number (Supplementary Fig. 2C) and area (Fig. 2I), and less circular spheroids (Fig. 2J) compared to the Wt PSA expressing MSK3 cells (Fig. 2H–J). Inactive mutant Ala[195] PSA and vector transfected cells behaved similarly in respect of all studied parameters in both cell lines (Fig. 2B–J). Thus, Thr[163] PSA expressing cells in spheroid models may potentially have a more invasive phenotype suggesting a dual role for the SNP in metastatic dissemination of cancer.

### Thr[163] PSA variant differentially modulates PCa cell behaviour in a bone metastasis model

Since PSA has been proposed to promote osteoblastic metastasis[27,28], a biomimetic in-vitro model of PCa metastasis to bone was developed and utilised. Here, stably transfected PC-3-furin-activable PSA cells were co-cultured with a 3D osteoblast-derived mineralised matrix (OBM) (Fig. 2K). OBM constructs were prepared from patient-derived osteoprogenitor cells and mineralised for 8 weeks[29]. Quantitative functional analysis of cancer cell attachment and proliferation on OBM were analysed (Fig. 2L, M). After an initial 12 h PC-3/OBM suspension co-culture (Fig. 2K), PC-3 cells from all groups (Wt PSA, Thr[163] PSA, Ala[195] PSA and vector) attached similarly to the OBM constructs (Fig. 2L). After a further 12 h and 24 h co-culture in serum free media, individual PC-3 cells attached to OBM constructs were measured for their shape factor and volume (Supplementary Fig. 3A). PC-3 cells displayed significant morphometric differences between groups. Similar to PC-3-vector cells, PC-3-Wt PSA expressing cells did not alter their shape, while a significantly reduced shape factor (*P* = 0.02) was observed for the PC-3-Thr[163] PSA cells (Supplementary Fig. 3A) and a spindle-like cell phenotype (as also observed in Supplementary Fig. 3B at 24 h), that may be associated with higher cellular plasticity for the PC-3-Thr[163] PSA cells.

PC-3 cells from all groups colonised the scaffold and images taken at 10 days (Supplementary Fig. 3B) appeared to demonstrate larger cellular volume for PC-3-Thr[163] PSA cells on the OBM, when compared to PC-3-Wt PSA cells, possibly owing to a differential substrate specificity for the Thr[163] PSA. Expression of Wt PSA reduced the proliferation (Fig. 2M) of PC-3 cells on OBM constructs compared to Vector cells

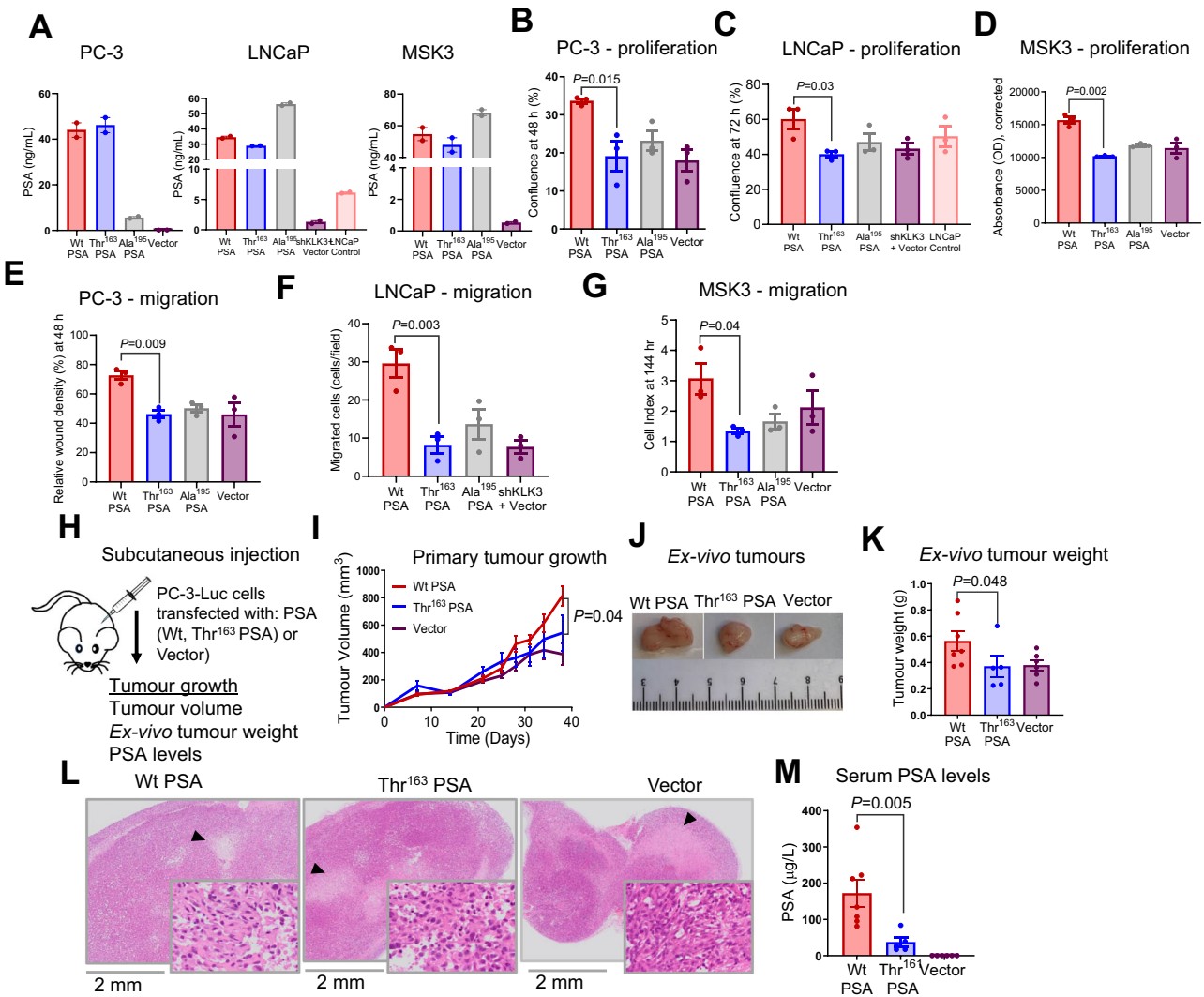

**Fig. 1 | Thr[163] PSA abolishes the effect of PSA on PC-3, LNCaP and patient-derived organoid MSK3 cell proliferation and migration and is associated with reduced growth of primary tumours in-vivo.** PC-3, LNCaP and patient-derived organoid MSK3 cells were transfected with furin-activable Wt PSA, Thr[163] PSA, Ala[195] PSA or control plasmid (vector). **A** Expression of PSA from engineered PSA constructs quantified by immunoassay (n = 2 independent experiments). **B, C** Proliferation rate (confluence %) measured in the IncuCyte live cell imaging system for PC-3 and LNCaP cells expressing PSA variants and vector control at 72 h (n = 3 independent experiments). **D** Proliferation of MSK3-PSA and vector control cells, measured by PrestoBlue cell viability assay at 144 h (n = 3 independent experiments). **E** Cell migration rate (relative wound density %) measured by the IncuCyte live cell imaging system for PC-3 cells expressing PSA variants compared to vector control at 48 h (n = 3 independent experiments). **F** Cell migration of PSA variants expressing LNCaP cells loaded in Boyden chambers at 48 h (n = 3 independent experiments). **G** Cell migration measured using the xCELLigence system for the PSA variant expressing MSK3 cells as compared to vector control (n = 3 independent experiments). **H** Preclinical subcutaneous xenograft tumour model of PC-3-Luc cells transfected with furin-activable Wt PSA, Thr[163] PSA or vector. **I** Mean volume of subcutaneous tumours throughout the experiment, based on caliper measurements (Wt: n = 7 mice, Thr[163]: n = 5 mice, Vec: n = 6 mice). **J** Representative photographs of resected subcutaneous tumours. **K** Scatter plot of post-mortem weight of subcutaneous tumours at day 38; horizontal line indicates mean value (Wt: n = 7 mice, Thr[163]: n = 5 mice, Vec: n = 6 mice). **L** H&E staining of resected subcutaneous tumours. **M** Serum concentration of total PSA at endpoint. All error bars represent mean ± SEM. Statistical analyses were determined by one-way ANOVA (**B-G, I**) or two-sided Student's t test (**K, M**). Source data are provided as a Source Data file.

(P = 0.02 for proliferation), supporting a tumour suppressive role for Wt PSA in the bone microenvironment. As compared to Wt PSA expressing cells, the PC-3-Thr[163] PSA cells displayed a more proliferative trend (Fig. 2M). Overall, PC-3-Ala[195] PSA and vector-PC-3 cells behaved similarly throughout all analyses and proliferated more rapidly than both PC-3-Wt PSA and PC-3-Thr[163] PSA cells (Fig. 2L, M, Supplementary Fig. 3). Our in-vitro data suggests that Thr[163] PSA expressing cells proliferate at a higher rate in the bone microenvironment in comparison to Wt PSA expressing cells.

### Thr[163] PSA increases metastasis in-vivo

To evaluate the context-dependent effect of the rs17632542 SNP in the tumour microenvironment in bone and ability to induce invasive

phenotype, the effects of the furin-activable PSA variants on bone metastasis in-vivo were investigated by intracardiac (left ventricular) injection of tumour cells for arterial blood dissemination (Fig. 2N). Based on bioluminescence imaging, the liver and kidneys were common sites of soft tissue metastasis, and the hind leg (tibia and femur) and mandible were frequent sites of bone metastasis. The livers (Supplementary Fig. 4A), hind legs (Fig. 2O–R, Supplementary Fig. 4B, C) and mandibles (Supplementary Fig. 4D, E) of mice injected with PC-3-Thr[163] PSA cells showed higher number of tumours, which also correlates with whole-body tumour load (Supplementary Fig. 4F) and serum PSA levels (Supplementary Fig. 4G) compared to those injected with Wt PSA or vector. All three transfected cell lines had the same baseline bioluminescence, as demonstrated by prior in-vitro

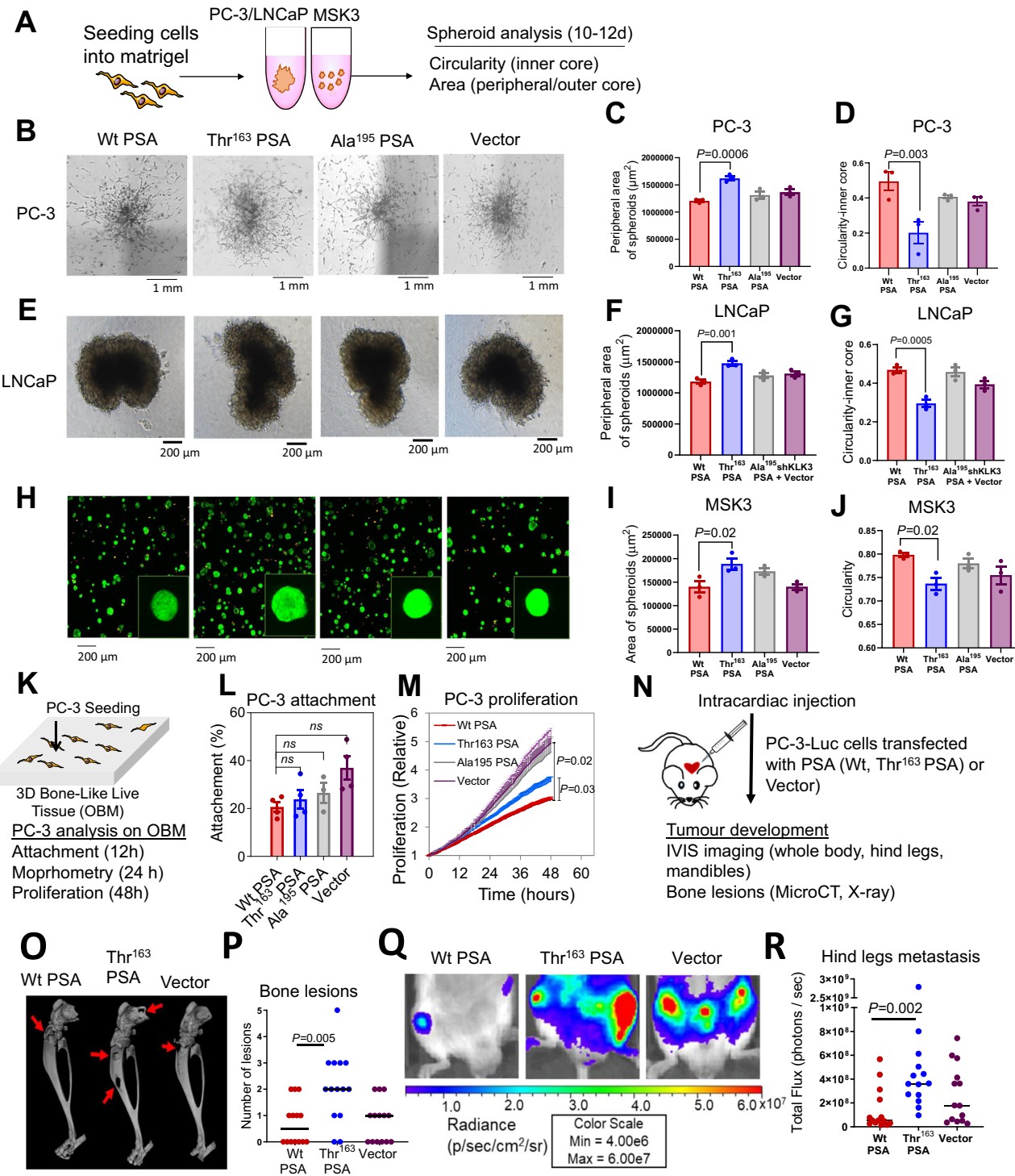

imaging (Supplementary Fig. 4H). Collectively, Thr[163] PSA was associated with highest metastatic tumour burden, including bone metastases, which is consistent with our observations for the behaviour of these cells in-vitro, suggestive of a relationship with the poor prognosis of the patients, carrying the rs17632542 SNP, encoding this PSA variant.

## Thr[163] PSA, has reduced activity towards peptide and protein substrates

Due to this conundrum for both protective and high PCa risk, we wanted to establish if the rs17632542 SNP leading to amino acid

substitution Ile to Thr at position 163 of the KLK3/PSA protein sequence, might affect the proteolytic activity of PSA.

Zymography of the recombinant PSA proteins on a casein gel indicated the Thr[163] PSA variant had lower activity than Wt PSA (Supplementary Fig. 5A). Additional proteolytic activity testing (Fig. 3A) with two peptide substrates, MeO-Suc-RPY-MCA and Mu-HSSKLQ-AMC (Fig. 3B), confirmed that the Thr[163] PSA had a lower proteolytic activity towards the fluorescent peptides compared to the Wt PSA protein variant and as expected, the mutant Ala[195] PSA control was inactive. The $K_{cat}$ for Thr[163] PSA was considerably lower than Wt PSA (Fig. 3B and Supplementary Fig. 5B).

**Fig. 2 | Thr[163] PSA increases cancer cell invasive ability and increases metastasis in-vivo. A** Schematic workflow of spheroid assay. **B** Representative brightfield microscope images (4× magnification) of 3D spheroids formed by transfected PC-3 cells after 10 days of culture. **C** Peripheral area (μm)² of invading cells outside the outer core. Also see Supplementary Fig. 2A. **D** Measure of invasiveness of the spheroid from 0 to 1 (1 = circular, least invasive; <1 = less circular spheroids). **E** Representative brightfield microscope images (4×) of LNCaP spheroids after 10 days of culture, **F** Peripheral area (μm)² and **G** circularity. **H** Representative fluorescent microscopy overlay images (10×) of transfected MSK3 cells at 10 days with a magnified view, stained with calcein-AM (live cells, green) and ethidium heterodimer (dead cells, orange) and spheroid, **I** Area (μm)² and **J** circularity (n = 3 independent experiments for (**B**–**J**)). Also see Supplementary Fig. 2B, C. **K** Schematic of a 3D osteoblast-derived bone matrix (OBM) co-culture with PC-3 cells. **L** Attachment of PC-3-mKO2-PSA cells to OBM constructs after 12 h co-culture. **M** PC-3 proliferation on OBM constructs. Also see Supplementary Fig. 3A, B. For

(**L**, **M**), n = 3 OBM groups from independent patient cells were made and included 2 technical replicates, 4–5 fields of view/replicate, for a total of 120–230 cells per condition. **N** Intracardiac injection of PC-3-Luc-PSA cells in mice (n = 7 mice/group). **O** Reconstructed 3D microCT images of tumour-bearing hind legs from representative mice of each group; red arrows showing areas of significant bone degradation, indicating presence of tumour. **P** Quantification of bone lesions per hind leg based on visual inspection of planar X-ray images; horizontal line indicates median value (n = 14 derived from two hind legs of 7 mice). **Q** Representative bioluminescence images of tumour-bearing hind legs of mice (week 4). **R** Scatter plots of tumour bioluminescence based on region of interest (ROI) drawn over individual hind legs (at week 4); horizontal line indicates median value (n = 14). Also see Supplementary Fig. 4. All error bars represent mean ± SEM; one-way ANOVA followed by Dunnett's multiple comparisons test (**C**, **D**, **F**, **G**, **I**, **J**), Dunn's multiple comparison test (**L**, **P**, **R**) or Games-Howell post hoc analysis (**M**). Source data are provided as a Source Data file.

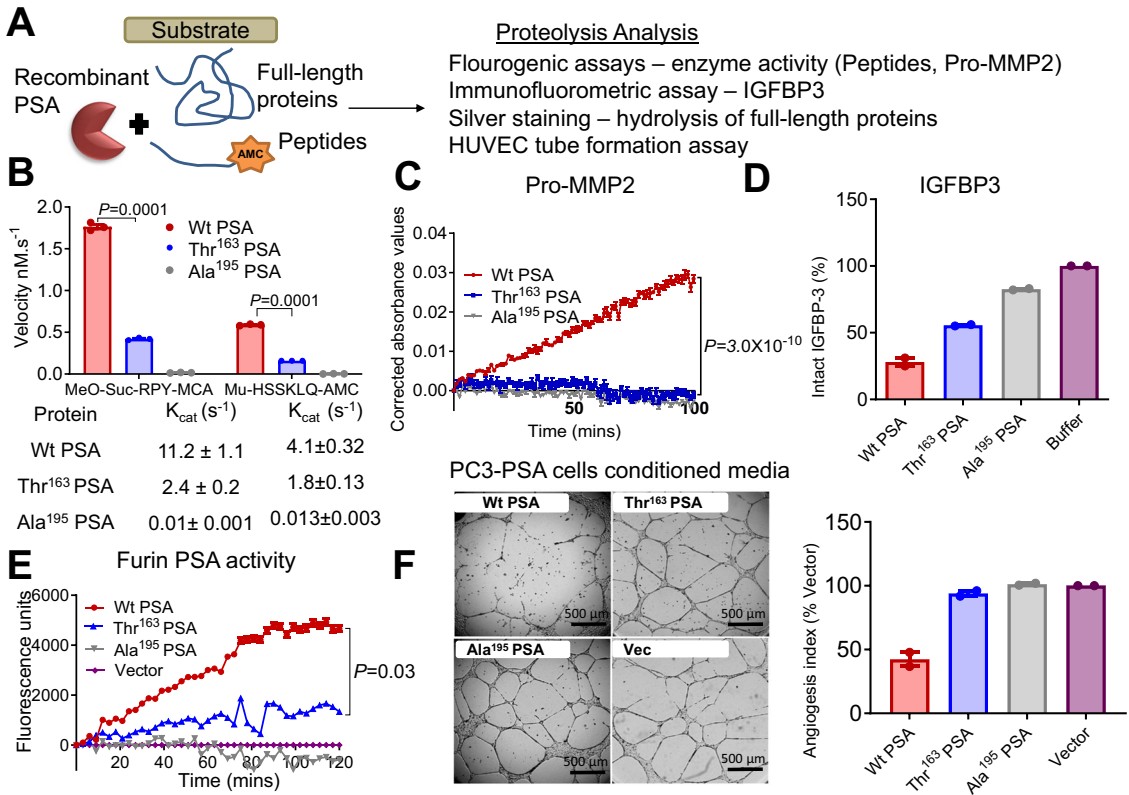

**Fig. 3 | Biochemical characterisation of the effect of the Thr[163] variant on PSA activity. A** Schematic for PSA proteolytic activity analysis. **B** Rate of hydrolysis by mature PSA proteins (Wt PSA, Thr[163] PSA, and catalytically inactive mutant control Ala[195] PSA, all at 0.1 μM) were compared using the peptide substrates MeO-Suc-RPY-MCA (10 μM) and Mu-HSSKLQ-AMC (1 μM) over 4 h at 37 °C. Proteolytic activity derived from assaying a constant amount of PSA with increasing concentration (0–250 mM) for these two substrates were used to estimate $K_{cat}$ values using nonlinear regression analysis in Graphpad Prism. (n = 3 independent experiments). Also see Supplementary Fig. 5B. **C** Time (mins) versus relative absorbance (OD) corrected to the substrate alone controls was plotted indicating the activity of pro-MMP2 (0.14 μM) when pre-incubated with PSA protein variants (Wt, Thr[163] and Ala[195] at 0.07 μM) and then the activity analysed with the chromogenic substrate (Ac-PLG-[2-mercapto-4-methyl-pentanoyl]-LG-OC₂H₅, 40 μM) for active MMP2 over 2 h. (n = 3 independent experiments). **D** Intact/total IGFBP-3 (2.2 μM) after 24 h incubation with PSA variants (0.25 μM) as shown relative to IGFBP-3 control

without added PSA. Also see Supplementary Fig. 5C. (n = 2 independent experiments). **E** Fluorescent activity observed for the furin generated active PSA captured from serum free conditioned media of furin-PSA overexpressing PC-3 cells (Wt, Thr[163], inactive mutant Ala[195] and vector) using the peptide substrate MeO-Suc-RPY-MCA (n = 3 independent experiments). **F** Inhibition of HUVEC tube formation on Matrigel by treatment of HUVECs with serum free conditioned media from the PC-3 cells overexpressing (Wt, Thr[163] and Ala[195] PSA) and Vector control. Scale bar is 500 μm. The graph to the right represents the effect of these PSA protein variants on HUVEC tube formation expressed as an angiogenesis index[49,65] is shown in relation to the control (n = 2 independent experiments). This is complemented by the same assay using recombinant PSA. Also see Supplementary Fig. 5D. All error bars represent mean ± SEM. Statistical analyses were determined by two-sided Student's t test (**B**) or one-way ANOVA followed by Dunn's multiple comparison test (**C**, **E**). Source data are provided as a Source Data file.

To further investigate the effect of the Thr[163] amino acid change on PSA function, we utilised several previously identified substrates of PSA[30,31]. Silver stain analysis after 22 h incubation of recombinant PSA-protein variants with the full-length protein substrates, semenogelin-1,

galectin-3, fibronectin, nidogen-1 and laminin α-4 demonstrated that the Thr[163] PSA had a lower proteolytic activity compared to the Wt PSA (Supplementary Fig. 5C). Furthermore, Wt PSA, but not Thr[163] PSA, can cleave pro-matrix metalloproteinase-2 (MMP2) leading to the

activation of zymogen and, thus, to an active MMP2 protease[32] (Fig. 3C). Similarly, the Thr[163] variant was less efficient in cleaving the substrate, IGFBP3 compared to Wt PSA (Fig. 3D). Together, this data, along with our substrate activity assays, demonstrate that the rs17632642 SNP reduces proteolytic activity of PSA but does not change its substrate specificity.

To confirm whether the PSA secreted (Fig. 1A) by the PC-3-PSA cells similarly exhibited a difference in proteolytic activity, PSA was captured by antibodies and activity was measured with the Meo-Suc-RPY-MCA substrate. Again, the measured PSA levels were similar for both the clones although the activity analysis showed the Wt PSA to be more active compared to the Thr[163] PSA (Fig. 3E) similar to our activity analysis with recombinant proteins (Fig. 3B).

## Thr[163] PSA variant has a reduced anti-angiogenic activity in comparison to Wt PSA

We hypothesised that the pro-metastatic activity of the furin-activable Thr[163] PSA observed in-vivo and altered biochemical activity may reflect the impact of the SNP on the anti-angiogenic role of PSA. Thus, a human umbilicial vein endothelial cells (HUVEC) endothelial tube formation assay was performed using conditioned media from the stable PC-3-PSA cells (overexpressing furin-activable either Wt PSA, Thr[163] PSA or Ala[195] PSA) and compared to conditioned media from control cells (PC-3-vector). HUVECs grown on top of Matrigel differentiated into tubular network structures during 16–20 h of incubation. Wt PSA containing media, when incubated with HUVEC cells, showed significant anti-angiogenic activity, decreasing the tube area to $35.2 \pm 2.5\%$ (mean $\pm$ SD, $P < 0.01$) compared to that of the cells treated with conditioned media from control cells (PC-3-vector). The Thr[163] PSA or inactive mutant Ala[195] PSA containing media did not significantly change the tube formation as compared to the control ($88.2 \pm 21.0\%$ and $108.4 \pm 30.9\%$ of the control, respectively, $P > 0.99$ for both) (Fig. 3F).

To confirm that the low anti-angiogenic activity, observed against HUVEC cells, in the conditioned media of PC-3-Thr[163] PSA cells was due to the impact of the secreted PSA, recombinant PSA variants (Wt PSA, Thr[163] PSA and Ala[195] PSA) expressed in, and purified from *Pichia pastoris* were utilised in tube formation assays (Supplementary Fig. 5D). A similar effect was observed emphasising that the antiangiogenic effect of PSA is dependent on a catalytically functional PSA and that Thr[163] PSA has a lower antiangiogenic activity compared to Wt PSA (Fig. 3F, Supplementary Fig. 5D).

## Thr[163] PSA variant has reduced ability to complex with (serum) protease inhibitors

We explored whether rs17632542 affects the complexing ability of PSA with serum inhibitors, thus affecting the f/t PSA that reflects both free PSA, which in blood circulation consists mostly of proteolytically inactive forms, and total immunoreactive PSA, i.e., both free PSA and PSA complexed to its predominant ligand in blood (α-1-antichymotrypsin/ACT/SERPINA3) (Fig. 4A). Silver stain analysis of recombinant PSA proteins with recombinant ACT verified a lower complexing ability of recombinant Thr[163] PSA compared to the Wt PSA as indicated by a lower intensity band of PSA-ACT complex at ~90 kDa compared to the Wt PSA (Fig. 4B). An additional band at ~70 kDa was observed which could be the PSA complexed with cleaved product of ACT (Fig. 4B). Since the complexing ability of PSA with inhibitors depend on its enzymatic activity, our results are in line with the lower activity observed for the recombinant Thr[163] PSA protein.

## The rs17632542 SNP [C] allele is associated with low total PSA levels and higher Free/Total PSA ratio compared to [T] allele

Recent studies, including ours, demonstrated that *KLK3/PSA* SNPs are significantly associated with serum PSA levels[11,22,24,33,34]. To confirm the allele specific effect, immunohistochemistry analysis was performed in patient tissue samples (TT = 10, CT = 10 and CC = 2) using an anti-PSA antibody to confirm the allele-dependent expression of PSA at the protein level. Reduced PSA ($P = 0.01$) protein levels were observed in tumour formalin fixed and paraffin-embedded (FFPE) slides of tumours from patients with the minor [C] allele compared to the [T] allele (Fig. 4C).

We analysed the genotype correlation with PSA levels in PCa cases and disease-free controls, since PSA levels may also be influenced by disease grade, stage and age of the individual. We thus assessed the genotype correlation in three independent sample sets. PCa cases, PRACTICAL consortium ($n = 31,770$; Fig. 4D); disease-free controls, the Malmö Diet and Cancer (MDC) Cohort ($n = 2458$; Fig. 4E) and The Västerbotten Intervention Project (VIP) Cohort ($n = 4810$; Fig. 4E) which all indicated lower total PSA (tPSA) levels for the rs17632542 SNP [C] allele.

Among men with modestly elevated PSA, risk assessment based on measuring both f/t PSA and tPSA is considered to have better predictive ability for PCa diagnosis compared to measuring tPSA alone[6,35]. To explore this further, we assessed the correlation of the rs17632542 [C] allele with f/t PSA ratio available for 958 PCa cases in five cohorts (IMPACT, PRAGGA, PROFILE, TAMPERE and ULM) of the PRACTICAL consortium sample set. In PCa cases, the f/t PSA ratio was $12.82 \pm 0.22\%$ for [TT] and $14.67 \pm 0.70\%$ for [CT] individuals (mean $\pm$ SEM, $P = 0.006$) and $21.5 \pm 9.5\%$ for individuals with [CC] genotypes (Fig. 4D). Similarly, the disease-free men with [CT] and [CC] genotype had significantly higher f/t PSA ratio in both MDC and VIP cohorts (Fig. 4E). The f/t PSA ratios were $32.89 \pm 0.18$ [TT], $38.32 \pm 0.64$ [CT] and $54.87 \pm 2.38$ [CC] (mean $\pm$ SEM, $P < 0.0001$) for VIP cohorts; and $34.11 \pm 0.27$ [TT], $38.89 \pm 0.72$ [CT] and $49.57 \pm 3.7$ (mean $\pm$ SEM, $P < 0.0001$) for the MDC cohort. This suggests that PSA in serum in men with [CT] and [CC] genotypes does not form complexes with protease inhibitors as efficiently as in men with [TT] genotype (Fig. 4D, E). Taken together, the [C] allele of the rs17632542 SNP may be associated with poor prognosis for PCa by its synergistic effects on protein expression and clinically measured serum PSA levels.

## *KLK3* rs17632542 SNP is associated with reduced PCa risk but increased metastasis and poor survival

We replicated the association between the rs17632542 SNP and PCa, with an odds-ratio (OR) = 0.70, 95% CI 0.67–0.73, ($P = 9.61 \times 10^{-69}$) for risk of any grade PCa diagnosis in a sample set of 49,941 PCa cases and 32,001 disease-free controls (Supplementary Table 1, Supplementary Table 2) using a custom high-density OncoArray. This association was similar after adjusting for family history (OR = 0.75, 95% CI 0.71–0.79, $P = 2.7 \times 10^{-26}$) and age of disease onset (OR = 0.75, 95% CI 0.71–0.79, $P = 5.2 \times 10^{-29}$) (Supplementary Table 2). The genotype data from this dataset for 46,939 PCa cases and 27,910 disease-free controls of European ancestry was combined with previously genotyped data for 32,255 PCa cases and 33,202 controls (from seven previous PCa GWAS imputed to 1KGP (2014 release)) of European ancestry. Estimated per-allele ORs for meta-analysis of 79,194 PCa cases and 61,112 disease free-controls were similar (OR = 0.74, 95% CI 0.72–0.76, $P = 6.69 \times 10^{-81}$) and the minor-allele frequency (MAF) of the [C] allele was 0.08. These results suggest that the *KLK3* rs17632542 SNP had a protective effect on PCa risk.

In a secondary analysis for survival within the OncoArray study samples, 37,316 cases were included. Of these 4629 died of PCa and 3456 died of other causes (PCa excluded as cause of death). Cases by carrier status were TT = 33,281, CT = 3909 and CC = 126. Despite the low minor allele numbers, the rs17632542 SNP was significantly associated with PCa specific mortality with a Hazard Ratio (HR) of 1.33, 95% CI = 1.24–1.45, $P < 0.001$ while for other causes of death HR = 1.08, 95% CI = 0.98–1.19, $P = 0.4$ (Fig. 4F). Validation in two independent longitudinal cohort studies of unscreened mid-life men also showed the SNP is associated with high PCa-related death; MDC (HR = 1.39, 95%

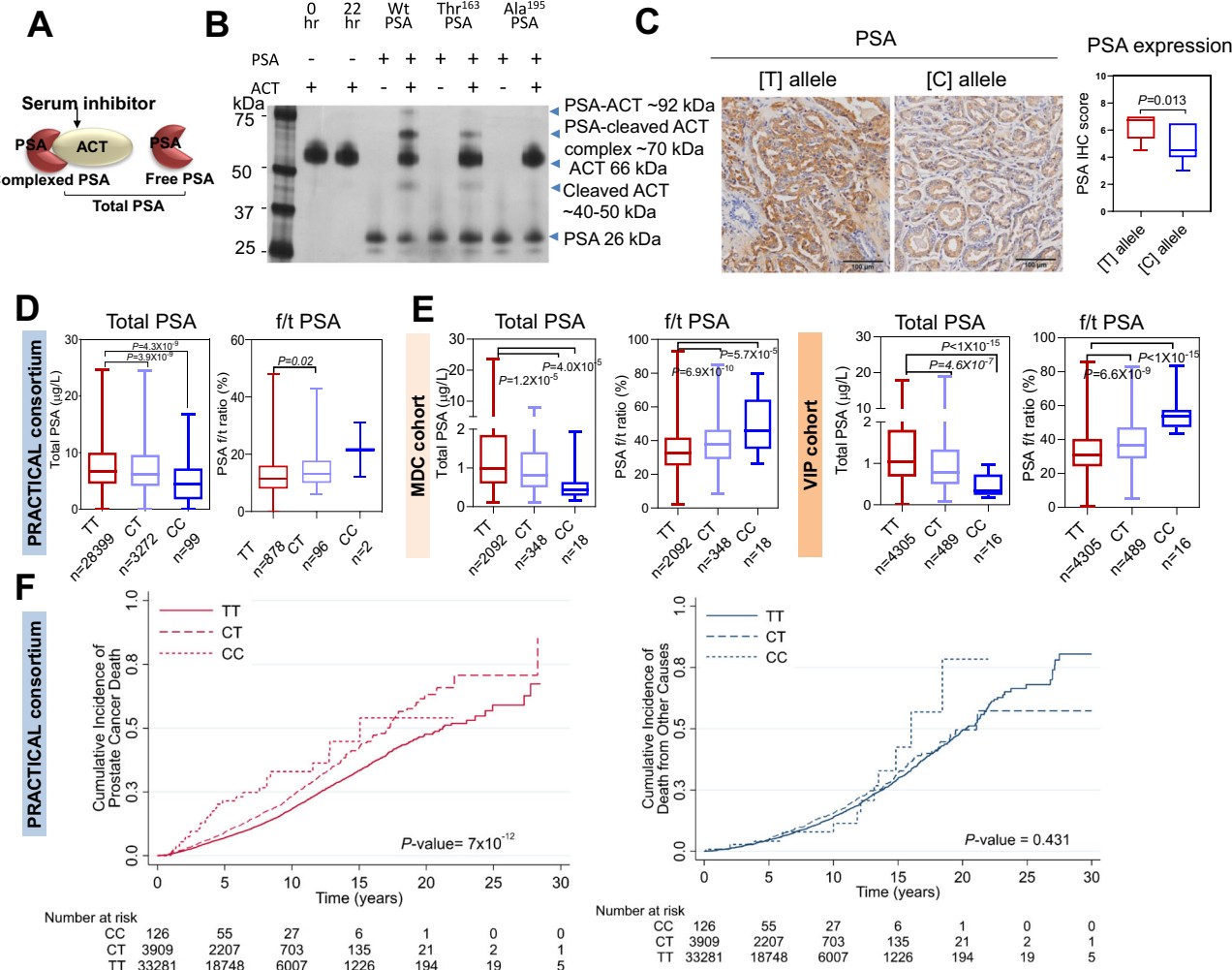

**Fig. 4 | rs17632442 SNP association with PSA levels and prostate cancer survival. A** PSA-inhibitor (ACT) complex, free and total PSA. **B** A representative silver stain analysis of recombinant wild type (Wt) and Thr[163] and Ala[195] PSA (0.1 μM) incubated with ACT (0.5 μM) at room temperature for 3 h before resolving on gel showed lower complexing potential of Thr[163] PSA with ACT compared to the Wt PSA. Inactive mutant Ala[195] does not complex with ACT (n = 3 independent experiments). **C** Representative immunohistochemical results for Gleason Grade 4 adenocarcinoma tissues, showing strong staining for PSA for the TT compared to the CC genotype. Graph on the right shows difference in PSA expression scores between [T] and [C] allele (CC: n = 2, CT: n = 10, TT: n = 10) for the immunohistochemical samples. The box plot centre line represents median, the boundaries represent interquartile range (IQR) and min and max are shown. **D, E** Genotype correlation of total PSA (tPSA) levels and f/t PSA ratio in prostate cancer cases (PRACTICAL consortium) and disease-free controls (MDC and VIP cohorts). **D** PRACTICAL consortium. n = 31,770; genotype status TT = 28,399, CT = 3272 and

CC = 99 for tPSA levels comparison. n = 976; genotype status TT = 878, CT = 96 and CC = 2 for f/t PSA ratio comparison. **E** MDC cohort with genotype status TT = 2092, CT = 348 and CC = 18; and VIP cohort with genotype status TT = 4305, CT = 489 and CC = 16. For box plots (**D, E**), median, inter-quartile range (IQR), min and max are shown. **F** Survival analysis for the rs17632542 SNP (c.536 T > C) in 37,316 cases of PRACTICAL consortium with follow-up on cause specific death. Of these, 4629 died of prostate cancer, 3,456 died of other causes. Cases by carrier status, TT = 33,281, CT = 3909 and CC = 126. The cumulative incidence of death from prostate cancer, Hazards ratio (HR) = 1.33, 95% CI = 1.24–1.45, P < 0.001 (left panel) and all causes other than prostate cancer, HR = 1.08, 95% CI = 0.98–1.19, P = 0.431 (right panel) are indicated. Number at risk are also indicated. All error bars represent mean ± SEM. Statistical analyses were determined by two-sided Student's t test (**C**) or one-way ANOVA followed by Dunn's multiple comparison test (**D, E**). Source data are provided as a Source Data file.

CI = 0.98–1.98, P = 0.06) and VIP (HR = 1.69, 95% CI = 1.07–2.65, P = 0.03) (Supplementary Fig. 6A, B).

Given the rs17632542 SNP is associated with PCa-specific mortality, we analysed if the SNP is associated with aggressive PCa susceptibility. Similar to the cumulative survival analysis, this SNP showed significant differences, but in opposite direction to our initial association analysis; for overall PCa risk, high risk (tumour stage T3/T4 or Gleason Score ≥8 or PSA > 20 ng/mL) vs low risk disease (tumour stage ≤T1, Gleason Score ≤6, PSA < 10) OR = 1.58, 95% CI 1.42–1.76, P = 1.23 × 10⁻¹⁷, high risk vs intermediate risk (Gleason Score=7, PSA = 10–20) OR = 1.42, 95% CI 1.33–1.51, P = 1.41 × 10⁻²⁶ and risk lethal vs controls OR = 1.33, 95% CI 1.16–1.51, P = 2.29 × 10⁻⁰⁵ (Supplementary Table 2). This association predicts whether the SNP

is associated with an increased risk of developing advanced stage PCa (tumour stage T3/T4), and therefore a poorer prognosis (Supplementary Table 2). We observed the correlation in a similar direction with risk of PCa death and metastasis-free survival in the VIP cohort. 1,667 PCa cases were selected for this analysis, of which 283 developed metastatic disease during >20 years follow-up. 286 cases were removed during quality control because their observation time was zero. Survival analysis showed rs17632542 is associated with metastasis-free survival time in VIP cohort (HR = 1.65, 95% CI = 1.03–2.62, P = 0.05) (Supplementary Fig. 6C). Together, these integrated analyses shows that the [C] allele of rs17632542 SNP is associated with increased risk for aggressive PCa susceptibility and PCa-specific mortality.

For OncoArray study samples where allele distribution with disease status is reported, the distribution of genotype frequency for rs17632542 SNP was calculated. The genotype frequency for this SNP varied with different disease stages as summarised in Supplementary Table 3. We observed higher frequency of the CT genotype at late cancer stage specifically in patients at both N1 (spread of tumour to lymph nodes) or M1 stage (distant metastasis) to be greater (0.15 and 0.13, respectively) compared with early-stage cases (0.10 for both N0 (no spread to lymph nodes) and M0 (no distant metastasis)). Thus, the rs17632542 minor [C] allele is protective against PCa risk overall in a large cohort, which is consistent with previous reports[21,34,36,37]. However, as we report here the SNP is associated with aggressive disease and higher risk of PCa death.

## Discussion

In recent years, the association of SNPs in the PSA encoding *KLK3* gene with PCa risk, PSA levels, or both has been debated, especially since these SNPs appear to influence PSA levels and thus may have influenced patient recruitment in these studies. Therefore, characterisation of the biological role may help define their risk association[38]. Here, we present an integrated study explaining the molecular and biochemical function of the protein isoform encoded by the rs17632542 SNP and the clinical implications underlying the *KLK3* PCa risk locus. We identified that the Thr[163] PSA variant reduces primary tumour growth but is also associated with a higher metastatic tumour burden. This dual risk association for the SNP was invariably supported by our association studies. In men carrying the rs17632542 [C] allele, we observed an overall lower risk of PCa but a higher incidence of PCa-specific death. Notably, the T > C substitution impacts the proteolytic activity of PSA with synergistic effects on serum f/t PSA levels that could lead to improved prediction of PCa clinical outcome.

The rs17632542 SNP is associated with reduced PCa risk[20–22,39,40]. However, the SNP association with PCa risk or PSA levels remains a conundrum. Thus, characterising the functional effects may provide more clues to uncover its role in prostate pathogenesis. Thr[163] PSA expression did not vigorously affect the cellular proliferation and migration of PC-3 and LNCaP cancer cells in controlled in-vitro cell-based assays. We verified these functional differences in an additional patient-derived organoid MSK3 cell line. These results are congruent with those that were obtained from our previous study that described Wt PSA over-expression to increase proliferation and migration of PC-3 cells[41], although an efficient expression system that releases catalytically active form of PSA (furin-activated) and an additional inactive Ala[195] PSA control was included in our study. While the shRNA targets *KLK3* 3′-UTR and exhibited a high knock-down efficiency (>80%) in LNCaP cells, we cannot exclude the possibility that it may not completely knockdown the expression of some of the KLK3 splice variants. In a previous study, we have shown miR-3162-5p has strong binding affinity to the T allele of *KLK3* rs1058205 miRSNP using reporter assays[42]. In our second study, using miR-3162-5p mimics, we demonstrated that, miR-3162-5p mediated knock-down of the *KLK3* gene, reduced protein levels of KLK3 and proliferation of LNCaP cells by additionally targeting other KLKs (KLK2, KLK4) and AR[43]. These two studies highlight the role of KLK3/PSA in the cellular function of PCa cells and validates our observation for lower proliferation and migration observed in the LNCaP-*KLK3* knockdown models. To determine Thr[163] PSA specific functional effects, we considered the PSA-deficient PC-3 overexpression models for our in-vivo analysis. The cells transfected with rs17632542 SNP behaved identical to vector transfected cells in our primary subcutaneous tumour in-vivo model, while Wt PSA expression promoted PC-3 tumour growth.

On the other hand, multicellular-spheroids, that mimic tumours in-vivo and a 3D bio-engineered osteoblast matrix bone model, allowed us to investigate the effect of the SNP on proliferation and shape factor of PCa cells. Additionally, an in-vivo model of metastatic cancer indicated the SNP to lead to the highest metastatic tumour burden, including bone metastasis, compared to Wt PSA. Overall, these analyses support a more proliferative and invasive capability of PCa cells expressing Thr[163] PSA. Consistently, PC-3 cells expressing Wt PSA showed an opposite trend supporting the notion that Wt PSA may have a tumour suppressive role on these cells in the metastatic tumour context specifically, the bone microenvironment. Our observations are in line with the anti-metastatic role of Wt PSA by hampering adhesion and invasive ability of PCa cells through prostate-derived extracellular matrix[44]. PSA is thought to mediate osteogenesis of mesenchymal stem cells via cadherin-Akt signalling[45] or affect bone homeostasis through increasing the bioavailability of osteoblastic growth factors such as IGF-1 and modulate genes involved in bone remodelling, such as RUNX-2, osteopontin and TGF-β[27]. PSA may also antagonize the Wnt pathway, by increasing Wnt inhibitory factors and reduce osteoblastic responses to PCa cells[27]. To what extent Thr[163] PSA can modulate these actions is not yet known but may suggest a differential substrate activity in comparison to Wt PSA, which may contribute to distinct cellular effects in PCa cells.

Treatment of HUVEC cells with Wt PSA reduced their angiogenic potential, but these cellular effects were observed to a lesser degree with Thr[163] PSA expressing cells or recombinant forms. Wt PSA exerts antiangiogenic activity in endothelial cell models in-vitro[46,47], however, recently it has been suggested to have a lymphangiogenic role as it activates VEGF-C and VEGF-D[48]. This supports previous observations of a dual role of PSA in tumour progression, promoting it by cleaving growth factors and ECM proteins or suppressing it by its anti-angiogenic potential and bone remodelling[49,50]. These studies, however, have only addressed the biochemical capability of PSA, not the bioavailability of PSA and its substrates in the tumour context. Thus, the biological significance of PSA antiangiogenic activity during progression of PCa is not well understood but suffice to say, that in the context of these cell-based models, that Thr[163] PSA does not possess the antiangiogenic activity of Wt PSA. However, we acknowledge the limitations in our in-vitro and in-vivo experimental models, which may not fully recapitulate the complexity of the tumour-microenvironment. For instance, our 3D models where we only used osteoblasts; and the use of normal endothelial HUVEC cells rather than tumour-associated endothelial cells to study the differences in anti-angiogenic effects. The relationship between PCa metastasis and death is complex, as upon diagnosis of metastasis, patients have a median survival of five additional years and much longer than other cancers. The development of castration-resistant metastases, or therapy-resistant metastases, contributes substantially to premature death, which prompted us to perform our assays in an androgen-dependent LNCaP and MSK-3 cells. Ascribing earlier mortality due to a non-proliferative, purely metastatic biology, with agnostic effects regarding AR signalling/inhibition, is a hypothesis that needs further experimentation. Nevertheless, in addition to the cell-based models utilised in this study, the strongest evidence is provided by our analysis in patient samples, indicating the rs17632542 SNP dual association with PCa risk and metastasis; and highlights the functional differences in the Thr[163] PSA expressing PCa cells compared to the Wt PSA, attributed through direct proteolysis of tissue-specific substrates or activation and perturbation of critical signalling pathways.

The proteolytic activity-dependent function of the SNP variant was also apparent in the lower ability to complex with the major PSA binding protein/inhibitor, ACT, a mechanism that requires active PSA[51]. This lower overall substrate binding affinity suggests a possible global structure perturbation that remotely affects the structure of the substrate binding site since the Thr[163] residue is outside the catalytic site[20]. Thus, disruption of PSA proteolytic activity by the Thr[163]-encoding allele may have a substantial impact on the involvement of PSA in PCa pathogenesis. For many years, it was debated as to whether PSA has a regulatory role in PCa biology or is just a surrogate biomarker for

assessing PCa progression. Our study invariably shows that PSA may have a multi-faceted role in the tumour context, by displaying a pro-tumourigenic role in localised tumours, but a suppressive role during tumour dissemination and metastasis. Overall, our findings are consistent with the context-dependent nature of *KLK3* gene function reported by others[52,53].

A further demonstration of the clinical relevance of rs17632542 SNP was provided by our results in PCa patient cohorts. The rs17632542 SNP is associated with lower serum PSA levels in our multi-cohort analyses and as reported previously[24,34,36] supporting a genetic basis for both tissue and circulating PSA levels. Percentage of fPSA contributes to modest diagnostic enhancements above and beyond tPSA alone among men in the "diagnostic gray zone"[54]. High %fPSA was also shown to be associated with worse survival outcome in patients with biochemical recurrence, indicating that fPSA may have role in progression to aggressive disease[55]. Recently, it has been reported that a different biology due to genetic variants underlies the high PCa-specific mortality observed in patients with Gleason Score of 9 to 10 and low PSA levels ≤ 4 ng/mL[56]. Two SNPs, located in introns 2 and 4 of the *KLK3* gene, and correlated with the rs17632542 SNP (r2 > 0.8), have been suggested to have potential regulatory effects on *KLK3* gene expression[20], but their effect on PSA levels has not been addressed to date. Our own recent study has shown that a second non-synonymous rs61752561 SNP in exon 3 of the *KLK3* gene has a potential role in PCa pathogenesis by addition of an extra-glycosylation site, changing protein stability and PSA activity and affects the clinically measured f/t PSA ratio[11]. Our study demonstrates that the rs17632542 SNP is associated with both higher ratios of f/t PSA due to its effect on reducing the ability to complex with inhibitors (PSA-ACT complexes), as well as lower levels of tPSA in blood which is expected due to the higher ratio of f/t PSA and much shorter clearance rate from blood compared to complexed PSA[57]. The lower PSA levels among the C-allele rs17632542 variant men are more likely prone to: 1) a negative detection bias as fewer of these men would be referred to prostate biopsies and; 2) due to this PCa-detection bias, more likely to be diagnosed with more advanced disease stages as their referral for a biopsy would be delayed due to a more modest PSA elevation and a higher ratio of free-to-total PSA. The high f/t PSA ratio may explain the protective effect of the C-allele rs17632542 variant in reference to risk of any grade PCa diagnosis.

The [C] allele of the rs17632542 SNP has been documented to be associated with lower PSA levels[20,34], reduced tumour volume[40] and reduced PCa risk[20,22,37,58]. This correlated with the risk association overall for the SNP in a large multicentre patient analysis herein, of which the major proportion of men contributing have low-grade disease. Survival analysis revealed poorer prognosis for the patients carrying the [C] allele in our multiple cohort-PRACTICAL study and two additional independent MDC and VIP cohorts. Our analysis shows the rs17632542 SNP [C] allele to be associated with PCa-specific death. We compared, high risk or fatal PCa and low risk disease and metastasis-incidence and found the [C] allele is associated with an increased risk of developing metastatic disease with the SNP allele more frequent in patients who have tumour spread to lymph nodes (N1) or distant metastasis (M1). The high frequency of the SNP in patients with aggressive cancer could also be attributed to their late detection owing to the low PSA levels.

Our study adds substantially to previous studies by indicating the potential for considering integration of SNPs with PSA into diagnostic pathways such as PSA polygenic score[18,25]. By applying genetic correction of PSA levels using 4 SNPs including the rs17632542 SNP, 6–7% of Icelandic men undergoing PSA screening, would have at least one PSA measurement reclassified with respect to whether they have to undergo prostate biopsy[34]. Using the same four PSA-SNPs it was suggested that, nearly 18–22% of unnecessary biopsies may be reduced by genetic correction[23]. While there is substantial evidence

demonstrating that the genetic background of individuals rather than SNPs within PSA can influence PSA levels, our study provides functional effects of germline variants on PCa tumourigenesis. Since the rs17632542 SNP is associated with poor survival, it is critical to carefully monitor men carrying either of the CT or CC genotypes as they may have aggressive cancer, without having abnormal total or f/t PSA values.

The current study has several important strengths. The identification of the rs17632542 SNP was based on a validation in several large-scale independent studies. To date, the relationship between PSA SNPs and PCa risk has remained obscure. We carefully applied gene over-expression strategies in three PCa cell lines (including a patient-derived organoid cell line) and clarified the functional and phenotypic relevance of the rs17632542 SNP with PCa pathogenesis making the association between the germline variant and PCa susceptibility, biologically plausible. The rs17632542 SNP, although associated with reduced PCa risk, is also associated with an aggressive phenotype and PCa-mortality, providing a rationale to develop a new personalised therapeutic strategy for PCa patients carrying the SNP allele. The rs17632542 SNP contributes to reduced serum PSA levels that may lead to detection bias during PSA screening leading to delayed diagnosis and treatment. Thus, these findings may allow better prognostic prediction, and in distinguishing a more lethal phenotype, to identify a high-risk group that need early treatment regimens. Combination of this SNP effect with other genetic variants reported recently[59–61] would also facilitate more accurate prediction of PCa risk. In our study we have observed the Wt PSA to have a protective role during PCa metastatic progression although the biology underlying the higher metastatic potential for the Thr[163] PSA still needs further investigation.

## Methods

### Mammalian cell culture

All procedures were performed in accordance with the QUT University Biosafety committee guidelines, University Human Research Ethics committee approvals (QUT#1500001082) and relevant ethical regulations for research. MSK3 cells were resourced through Dr Ian Vela, Queensland University of Technology; patients provided informed consent and samples were acquired under MSKCC IRB-approved protocols # 06-107 and 12-001. The isolation of HUVECs was approved by the Institutional Review Board of Helsinki University Central Hospital (application number 112/E9/06). All participants gave a written informed consent for the use of the cells and did not receive any compensation. Human mesenchymal osteoprogenitor cells were obtained under informed consent from male patients undergoing hip or knee replacement surgery (QUT ethics approval number 1400001024)[62].

The androgen-independent bone metastasis-derived human PCa cell line, PC-3, androgen-dependent human metastatic PCa cell line, LNCaP and HUVECs (for studying PSA variants secreted into conditioned media by PC-3-PSA cells), were purchased from the American Type Culture Collection. PC-3 cells were maintained in RPMI-1640 medium supplemented with 5% Fetal Bovine Serum (FBS) and passaged using Versene (Invitro Technologies) in an atmosphere of 5% $CO_2$ and 99% relative humidity at 37 °C. Patient-derived organoid MSK3 cells, a mucinous adenocarcinoma isolated from a retro-peritoneal lymph node generated at Memorial Sloan-Kettering Cancer Center (MSKCC)[26]. MSK3 cells were maintained in a serum-free conditioned prostate culture medium[63] and passaged using TrypLE (Invitro Technologies). HUVECs isolated from umbilical veins[64,65] were cultured in endothelial cell growth medium (PromoCell). All cell lines were tested for mycoplasma. With respect to their genotype status for the rs17632542 SNP, PC-3 is heterozygous CT genotype, while LNCaP and MSK3 are homozygous TT genotype (data obtained through RNA sequencing data of the native cell lines).

Human mesenchymal osteoprogenitor cells were isolated from bone tissue and cultured in growth media (GM), containing alpha-Modified Eagle Medium (alpha-MEM), supplemented with 10% FBS, 100 U/mL penicillin and 100 μg/mL streptomycin (all from Thermo-Fisher Scientific). Cells were used at passages 3–5 and were mycoplasma free.

### Construction of plasmids for PSA variant expression

To ensure the activation of the expressed PSA, the expression constructs were engineered by changing the region encoding the pro-domain (APLILSR) of the PSA sequence to one encoding a furin recognition sequence (APLRLRR)[66]. The pcDNA3.1-PSA vectors encoding furin activatable Wt PSA were generated by cloning this engineered PSA sequence into the EcoRI and XhoI digested pcDNA3.1 vector[11]. Site-directed mutagenesis was performed to create the SNP allele isoform, Thr[163] PSA, and catalytically inactive (Ala[195] PSA) PSA isoform using mutagenic primers (Supplementary Table 4). Mutated PSA sequences were confirmed by Sanger sequencing using T7 and BGH primers (Supplementary Table 4).

To generate lentiviral PSA overexpression vectors, pCDNA3.1-PSA (Wt and Thr[163]) vectors generated above were used as template and amplified using attB overhangs and subsequently cloned into a pLEX307-Puro overexpression plasmid. The pLEX307-GFP plasmid was used as a control. For PSA knockdown, pLV-mCherry-U6 > hKLK3 plasmid (shRNA#1- *GTGTTTCTTAAATGGTGTAAT*) and pLV-mCherry-U6-scramble vector (Vectorbuilder) were utilized.

To generate luciferase-labelled PSA-expressing PCa cells for in-vivo models, PC-3 cells were transfected via a lentiviral vector-base method. cDNA encoding luciferase protein from a pGL4.10-luc2 plasmid (Promega, Sydney, Australia) was cloned into a pLenti CMV Hygro DEST vector (Addgene, Cambridge, MA) using Gateway LR recombination cloning technology (Life Technologies). Cells stably infected with the luciferase construct were selected in hygromycin (1 mg/mL) containing medium.

### Cell models for expressing Wt and PSA variants

PSA constructs generated as described above were transfected into PC-3 (do not express PSA), LNCaP (high PSA expression) and patient-derived organoid MSK3 (low PSA expression) cells (50,000 cells) seeded into 24 well plates using the FuGENE® transfection reagent (Promega) according to the manufacturer's instructions (1:3 ratio of DNA to lipid used). PSA expression levels by the PC-3/MSK3-PSA polyclonal populations were tested by qRT-PCR (PSA primers: Supplementary Table 4) and Western blot analysis using an anti-PSA antibody (Dako, #A0562) before subsequent characterisation below.

For lentiviral viral transduction, lentiviral particles were generated in HEK293T host cells transfected with FuGENE® transfection reagent (Promega) and pLEX307-fPSA/Vec plasmids generated above for overexpression of PSA or pLV-mCherry-U6 > hKLK3 for PSA knockdown. The pCMV-8.2 R lentiviral packaging plasmids and pCMV-VSVG were kindly provided by Dr Brett Hollier (Queensland University of Technology, Australia). Virus particles were collected after 48 h of transfection, filtered through a 45 μm filter and added to the cell lines for subsequent selection by Fluorescence Assisted Cell Sorting (FACS) using an Astrios cell sorter (Beckman Coulter, Australia) for mCherry (mCherry-High and mCherry-Low) or antibiotics (1 μg/mL puromycin for pLEX307-puro). In PSA knock-down cells, the knock-down efficiency is confirmed by qRT-PCR and for re-expression of PSA variants in PSA-knockdown cells.

For evaluating the morphological effect of the PSA variants on bone scaffolds, the PC-3-PSA cells were re-transfected with the pLEX307-mKO2 lentiviral plasmid (a kind gift by Dr Sally Stephenson, Queensland University of Technology, Australia), sorted by FACS for mKO2-High and mKO2-Low, and verified for PSA expression prior to use.

### qRT-PCR for PSA expression analysis

Total RNA was extracted from PC-3, LNCaP and MSK3 PSA over-expressing and vector cells using the Isolate II RNA mini kit (Bioline, Australia) according to the manufacturer's instructions. One μg of RNA was reverse transcribed using SuperScript III reverse transcriptase (Invitrogen) and amplified using the SYBR Green PCR Master Mix (Applied Biosystems®). The primer sequences are listed in Supplementary Table 4. Relative expression levels of the target genes were determined by the comparative $C_T$ ($\Delta\Delta C_T$) method[67].

### Immunofluorometric assay for free and total PSA

The secretion of PSA by the PCa cell lines was determined with a dual-label DELFIA immunofluormetric assay (PROSTATUS™ PSA Free/Total PSA from Perkin Elmer, Australia) or Total PSA ELISA kit (Aviva systems biology, San Diego). Briefly, the PSA in the conditioned media was captured to the immobilised anti-PSA antibody and the free to total PSA ratio or total PSA were calculated[68,69].

### Analysis of cell proliferation and migration of PSA variant expressing cells

For PC-3 and LNCaP cell proliferation analysis, 5000 PSA variant transfected cells were seeded overnight in 96-well flat-bottomed plates and monitored in the IncuCyte live cell imaging system (Essen Biosciences) in serum free conditions or media containing 2% FBS, respectively, over 48–72 h. To account for PCa cell growth as aggregates, proliferation for PSA variant transfected patient-derived organoid MSK3 cells was assessed using PrestoBlue reagent (Invitrogen, Australia).

For the PC-3 cell migration assay, $3 \times 10^4$ cells were plated per well in a 96-well ImageLock plates (Essen Biosciences) and incubated overnight at 37 °C (Sigma Aldrich) to form a confluent monolayer of cells. The cells were pre-treated with Mitomycin-C (at 10 μg/mL) for 2 h before a scratch was made using a 96-pin WoundMaker™ (Essen Biosciences). To validate the differences in migration using an alternative approach, serum-free media containing $1 \times 10^5$ LNCaP cells were loaded on to the top chamber in Boyden chamber plates (24-well) and media containing 10% FBS was added in the bottom chamber. Cells were allowed to migrate for 18 h across the membrane. Cells were fixed in 100% methanol for 2 min and stained with 1% crystal violet for 2 min, followed by 2X washes in PBS. Using a sterile cotton swab, the non-migratory cells in the upper chamber were removed and the average number of migrated cells from three fields/well were counted with a EVOS FL microscope (Thermo Fisher Scientific). Migration for the patient-derived organoid MSK3 cells were assessed using the xCELLigence system which is based on the principle of the Boyden Chamber assay, according to the manufacturer's instructions (Roche). Briefly, $5 \times 10^4$ cells were plated per well and cell index/time was derived using the RTCA software. At least three technical replicates per group were included. In total three biological replicates were performed.

### In-vivo mice models

**Animal ethics statement.** All studies were performed in accordance with guidelines of the Animal Ethics Committees of The University of Queensland (AEC number: 091/17) and Queensland University of Technology, and the Australian Code for the Care and Use of Animals for Scientific Purposes.

Male NOD SCID gamma (NSG) mice, 5–6 weeks old ($n = 7$ mice/group), were sourced from the Australian Resources Centre (ARC; Australia). We used age-matched male mice in our studies to model PCa. All mice were maintained at the Biological Resources Facility at the Translational Research Institute, Woolloongabba, QLD.

**In-vivo tumorigenesis studies.** Subcutaneous implantation of $1 \times 10^6$ PC-3-Luc-furin activatable PSA (Wt, Thr[163])/Vec cells in PBS was performed on the right flank of 5–6 weeks old male NSG mice in 100 μL

volume. The tumours were measured using electronic calipers every 2–3 days and tumour volume calculated from the formula for the volume of an elipse: $V = \pi/6(d_1.d_2)^{3/2}$, where $d_1$ and $d_2$ are two perpendicular tumour dimensions. Tumour volumes of 1000 mm³ was considered as the humane endpoint in accordance with the ethics approval (AEC number: 091/17) and was not exceeded. In the metastasis model, $2 \times 10^5$ PC-3-Luc-furin activatable PSA (Wt, Thr[163])/Vec cells were injected into the left ventricle of 5–6 weeks old male NSG mice mice for arterial blood dissemination, a technical procedure guided by a small animal ultrasound imaging station (Vevo 2100, Visualsonics, Canada)[70].

**Tumour bioluminescence imaging.** Tumour development was monitored by weekly bioluminescence imaging using an IVIS Spectrum (Perkin Elmer, USA). For in-vivo imaging, mice were injected intraperitoneally with D-luciferin diluted in PBS (15 mg/mL stock) at 150 mg/kg, anaesthetised and imaged until tumour bioluminescence plateaued. Bioluminescence was analysed using Living Imagine software (Xenogen, CA, USA). The total flux in photons/second (p/s) within each defined region of interest (ROI) provides a surrogate of tumour burden. For in-vitro imaging, bioluminescent cells were seeded at 50,000 cells/well down to 50 cells/well (2-fold serial dilution) in 96-well plates. D-luciferin (Perkin Elmer, USA) was added to each well (final concentration was 150 μg/mL of media) 3–5 min prior to imaging.

**High resolution microCT (ex-vivo).** High resolution microCT imaging was performed using a Skyscan 1272 (version, 1.1.19; Bruker, Belgium). Mouse leg specimens were fixed in 10% neutral-buffered formalin for 48 h, stored in 70% ethanol, then wrapped in moist tissue paper and transferred into 5 mL cylindrical plastic tubes for imaging. The scanning parameters were: 70 kV X-ray voltage, 142 uA current, 600 ms exposure time, 19.8 μm isotropic voxel size, 0.5° rotation step (360° imaging), 2 frame averaging, 4 × 4 binning, and 0.5 mm Al filter. The datasets were reconstructed with NRecon (Bruker) and InstaRecon (University of Illinois, USA) software using cone beam reconstruction (Feldkamp) algorithm and the following corrections applied: ring artefact reduction, beam hardening, and post-alignment. CT analysis was performed using CTAn software version (Bruker), and 3D visualisations of legs generated using CTVox software (Bruker).

**X-ray radiography (ex-vivo).** Post-mortem X-ray imaging of resected mouse hind leg bones was performed using a Faxitron Ultrafocus digital X-ray system (Faxitron Bioptics, USA).

**Histologic analysis of mouse tissues.** Subcutaneous tumours and tumour bearing tissues for metastasis models harvested ex-vivo were fixed in 4% paraformaldehyde. Histologic analysis was performed for confirming the presence of tumour cells in specific organs and mice hind legs at the end of the experiment. Bone specimens were decalcified in 10% EDTA in PBS for two weeks and the decalcified bones were separated and embedded in paraffin blocks. Serial sections of both subcutaneous tumours and mice legs with metastatic lesions were stained with hematoxylin and eosin (H&E).

### 3D spheroid cell models and morphological analyses

To monitor changes in invasiveness and tumour-specific differentiation patterns, PC-3 and LNCaP cells transfected with furin-activatable PSA variants were plated at 1000 cells/well on an Ultra-low Cluster 96 well plate (Sigma Aldrich, Australia) in low FBS (2%) containing RPMI media. After 4 days, 100 μL of phenol-red free growth-factor reduced Matrigel® matrix (Corning, USA) (10 mg/mL) was added to each well, topped up with 100 μL media (2% FBS) after 1 h and incubated for 10 days. Spheroids were imaged using a Nikon spinning disc confocal microscope or Nikon ECLIPSE Ts2R using 4× objective.

To generate MSK3 spheroids, MSK3 cells transfected with furin-activatable PSA variants were embedded between two matrigel layers (4000 cells/well). After 10 days in 3D culture, live/dead staining was performed using Calcein AM live cell dye and ethidium homodimer (both from ThermoFisher Scientific, Australia), respectively. Stacks of spheroid images of MSK3 cells were taken with an INCell Analyzer 6500 HS high content analysis system (GE Healthcare Life Sciences, Australia). Digital analysis was acquired for the images on spheroid - number, size, morphology (circularity/compactness) and viability of spheroids (live - Calcein/green and dead -heterodimer/orange staining) were quantified using a custom analysis pipeline in the StrataQuest™ image cytometry software (TissueGnostics, Vienna, Austria) to automate the quantitative analyses of spheroids. For PC-3 and LNCaP spheroids, live/dead cell staining could not be performed since we observed a high background of calcein staining the matrigel.

To perform image cytometry, several analysis engines were defined in the image analysis environment, StrataQuest, to process the original Tiff images in a pipeline process. A grey channel image was generated from the original image. The background was removed from the grey image to correct for illumination artefacts using a set of engines to locally reduce the background. Then a threshold was applied to the images for the detection of positive objects and a density image was generated. The high-density area was split into two parts – the inner core and outer core based on intensity. The periphery is set as the complement of the two parts. Three areas were generated as shown in Supplementary Fig. 2A, with the green contour overlay highlighting the outer core and the orange contour overlay indicating the area of the inner core. The blue contour overlay around the spheroids contains detectable cells in the periphery. Finally, after the recorded images of single PC-3 and LNCaP stellate spheroids per well were segmented, several measurements were performed. The read-out parameters include the circularity of the dense central/inner core of each spheroid, the area of the outer core and the area of the peripheral invading cells, to indicate invasive ability. Manual correction was performed to remove artefacts, where necessary, to assure data consistency.

For MSK3 patient-derived organoids that formed multiple circular spheroids, circularity of the whole spheroids and additional properties such as live/dead cell staining, number of spheroids and maximum intensity projections created from z-stacks were determined. The original Tiff images, in sets of two 16-bit grey scale images, one each for the green Calcein and the orange ethidium homo-dimer markers were used. Dead cells were detected using a combination of two detection engines. First a detection of dotlike structures, with high intensity in the centre and lower (gradient) intensity around the centre. The second step was a detection of specific stained areas / marker positive cells using an intensity threshold operation. Both segmentation masks were merged for a final detection of the dead cells. Spheroids were detected based on a double threshold on intensity and area. (Supplementary Fig. 2B).

Statistics were generated automatically based on total event count as measure of spheroid number, count, and mean intensity for live and dead cells within the spheroid and event area for spheroid area. Manual correction for automatic cell detection was performed for single live/dead cells, where necessary, to compensate for air bubbles and other erratic background patterns. At least two technical replicates per group were included. In total three biological replicates were performed.

### Co-culture models of osteoblasts with PC-3- PSA -mKO2 expressing cells

**Scaffold Fabrication.** Microfibre scaffolds made of medical-grade polycaprolactone (mPCL) were produced by melt-electrospinning with an in house-built equipment and protocol[29]. Briefly, mPCL loaded into a Leur-Lock plastic syringe (Nordson EFD, Australia) was pre-heated at

60 °C and fitted with a tapered needle and set in the MEW block heaters set at 74 °C and 85 °C for syringe and needle block heaters, respectively. After 2 h, extrusion pressure (2.2 bar) was regulated and the working distance between needle and aluminium collector is maintained at 9 mm. After loading a G-code software in the March 3 software (Artsoft, USA), the voltage was increased (101.1 kV) and print was initiated for 2.5 days until completion. Final scaffolds were 12 × 12 × 0.4 mm in size, with a 3D interconnected structure and 150 μm pore size. Scaffolds were coated with calcium phosphate to promote cell adhesion and osteogenic differentiation[71].

**Scaffold Culture.** After sterilization with 100% Ethanol and UV radiation (20 min both sides), mPCL scaffolds were seeded with osteoprogenitor cells (800,000 cells/scaffold) in a 5 μL drop in the centre of the scaffolds. After attachment (4 h), scaffolds were cultured in growth media (GM) (MEM-Alpha with 10% FBS and 1% Penicillin/Streptomycin) until they reached 3D confluency within the scaffold. Media was then changed to osteogenic media (OM), containing GM + 10 nM β-glycerophosphate, 0.17 nM ascorbic acid, 100 nM dexamethasone (all supplied from Sigma-Aldrich, Australia) and scaffolds were cultured for 8 weeks until mineralization occurred. Media change was performed 2 times a week with fresh OM made weekly. The final osteoblasts/scaffold constructs are referred to as 'OBM constructs' and displayed relevant bone characteristics (collagen deposit, mineralization)[72].

**OBM Co-Culture with PC-3-PSA-mKO2 Cells.** Once mineralised, OBM constructs were washed in serum-free RPMI media 3 times. Biopsy punches (5 mm) were made from the constructs and placed in a 24-well plate prior to seeding of PC-3 cells overexpressing Wt, Thr[163], Ala[195] PSA or Vector, re-transfected with pLEX307-mKO2. PC-3-PSA-mKO2 cell solutions were prepared in serum-free RPMI at a concentration of 50,000 cells/mL. 500 μL was seeded on the scaffolds (25,000 cells total/well) and incubated (37 °C, 5% CO₂) overnight on a shaking platform. Upon PC-3-mKO2 cell attachment to OBM constructs (12 h), cell suspensions were removed and counted to determine the degree of PC-3 attachment to OBM, and the constructs were washed 3 times with serum-free RPMI. PC-3/OBM co-culture (CC) constructs for morphometry were then placed in new 24 well-plates and cultured for a further 12 h in serum-free RPMI. CC constructs were then washed 3 times in serum-free RPMI and fixed in 4% paraformaldehyde for 3 h, followed by 3 washes in PBS and stored at 4 °C until staining. Quantitative functional analysis of cancer cell attachment, morphometry, and proliferation on OBM has been established previously for PCa cell lines[73], and was applied here. Briefly, for morphometry, automated surface statistics were computed from z-stacks in Imaris and proliferation, live cell image series were analysed by ImageJ software. For proliferation, some CC constructs were used for live cell imaging for a further 48 h in serum-free conditions, after the initial 12 h attachment. For long-term cultures, CC constructs used for live imaging experiments were further cultured in 5% FBS-RPMI up to 10 days. While in culture, CC constructs were monitored with an Olympus BX60 microscope using a CY3 (red) filter to identify PC-3-mKO2 cells on OBM, and bright field for general topography. After 10 days, CC constructs were washed 3 times in serum-free RPMI and fixed in 4% paraformaldehyde for 3 h, followed by 3 washes in PBS and stored at 4 °C until staining.

**Immunofluorescence staining.** PC-3/OBM constructs were stained by DAPI (5 mg/mL) for nuclei staining and Alexa Fluor Phalloidin 488 for actin staining (0.8 U/mL), (ThermoFisher Scientific, Australia), diluted in 0.5% Bovine Serum Albumin (BSA) in PBS (Sigma-Aldrich, Australia). Constructs were incubated for 45 min at room temperature with the staining solution, rinsed 3 times in PBS (10 min per rinse) on a shaking platform. Constructs were transferred to 2 mL Eppendorf tubes supplemented with fresh PBS and stored at 4 °C until analysis.

**Fixed Imaging.** PC-3/OBM constructs were imaged for morphometry (fixed after 1 day co-culture) and for overall morphology (fixed after 10 days co-culture), on a Nikon Spectral Spinning Disc Confocal microscope (X-1 Yokogawa spinning disc with Borealis modification) fitted with a 10X PlanApo objective, using green (FITC, excitation at 488 nm, laser power at 72%, exposure time 300 ms, Gain 1.5×), red (CY3, excitation at 561 nm, laser power at 73%, exposure time 400 ms, Gain 1.5×) and blue (DAPI, excitation at 405 nm, laser power at 54%, exposure time 100 ms, Gain 1.5×) filter sets. Z-stacks were obtained from 51 images taken every 1 μm over a 50 μm thickness, comprising the PCa cell layer on top of the OBM. Four different fields of view were collected for morphometry analysis per CC construct and 2 constructs/condition were used.

**Live cell imaging.** Live PC-3/OBM constructs were placed in a 24-well plate in serum-free conditions (500 μL) and secured down using Teflon rings. An Olympus Live Cell microscope was used to record videos of cells for 48 h. Images were taken every 20 min (4X objective) using CY3 (red) to identify PC-3-mKO2 cells moving on OBM, and bright field channels for general topography. Videos were reconstructed from images (145 frames in total). An average of 8 fields of view were recorded per CC construct and 2 constructs/condition were used.

**Image analysis.** For morphometric and migration studies, images were analysed using Imaris imaging analysis software (Version 8.4.1, Bitplane AG, Zurich, Switzerland). For morphometric analysis (cellular volume and sphericity), automated surface statistics were computed from Z-stacks (algorithm parameters: Surface area detail 1 μm, Threshold: Automatic, Diameter 11 μm, Quality filter: Automatic) for at least 100 cells per group. For migration analysis (speed), automated spots statistics were computed from live cell imaging series (algorithm parameters: Estimated cell diameter 18 μm, Intensity filter 30-230, Max distance jumps 20 μm, Max gap size 5) for 120–230 cells/tracks per group. For proliferation studies, live cell image series were analysed using ImageJ software (Version 1.51 h, Rasband, W.S., ImageJ, U.S. National Institutes of Health, Bethesda, Maryland, USA). In brief, the area occupied by PC-3 cells at each time point was measured by setting a high intensity threshold for the mKO2 (red) signal and using the area measurement function of ImageJ. An average of 8 fields of view were recorded per CC construct and 2 constructs/condition were used.

**Production of recombinant active PSA**
For recombinant protein overexpression in *Pichia pastoris*, *KLK3* cDNA (NCBI RefSeq: NM_001648.2) cloned in the pCDNA3.1/V5-6His vector[41] was engineered to include a pre-signal sequence for secretion in *Pichia pastoris* and then cloned into the pPIC9K vector (Invitrogen) conferring a N-terminal enterokinase and hexahistidine (6His) tag. Single point mutations were generated using mutagenic primers to generate the Ile163Thr (Thr[163] PSA) and Ser195Ala (Ala[195] PSA) substitutions followed by expression in *Pichia pastoris* GS115 cells.

Transformants expressing high levels of each of the protein variants were chosen for larger scale expression and purification by cation exchange chromatography and the purified proteins were further subjected to enterokinase (EK) digestion and purified by cation exchange chromatography.

**In-vitro enzymatic assay for the secreted PSA and variants**
Secreted PSA in conditioned media was captured on a 96-well plate by a PSA specific antibody (PROSTATUS™ PSA Free/Total PSA from Perkin Elmer, Australia) as described above. The activity of the captured PSA specific was determined by the addition of a fluorescent peptide substrate (MeO-Suc-RPY-MCA, 1 μM/well/200 μL) diluted in TBST assay buffer (0.1 M Tris base pH 7.8, 0.15 M NaCl, 10 mM CaCl₂, 0.005% Triton X-100). The plate was incubated with slow shaking at 37 °C and fluorescence was measured at 355 nM (excitation) and 460 nm

(emission) every 3 mins for approximately 4 h. Three technical replicates per group were included. In total three biological replicates were performed.

## PSA activity assays with peptide and protein substrates
**Determination of PSA enzyme activity.** The enzymatic activity of the recombinant PSA variants was measured using two fluorescent peptides (MeO-Suc-RPY-MCA[74] (Peptides International) and Mu-HSSKLQ-AMC[75] (Sigma Aldrich, Australia). Fluorogenic assays were performed in 384-well microplates (Corning). PSA proteins ($0.1\,\mu M$) were incubated with $1–10\,\mu M$ fluorogenic substrates in 50 mM TBST buffer for the MeO-Suc-RPY-MCA substrate or TBS (0.1 M Tris base pH 7.8, 0.15 M NaCl, 10 mM $CaCl_2$) with 0.1% BSA for the Mu-HSSKLQ-AMC substrate. The plates were incubated for 4 h at 37 °C and fluorescence was measured at 355/460 nm (excitation/emission) with a POLARstar Omega Plate Reader Spectrophotometer (BMG labtech). Three technical replicates per group were included in three independent biological replicates.

The $V_{max}$ (maximum rate of reaction), $K_m$ (Michaelis constant) and $K_{cat}$ (catalytic rate constant) were determined for PSA with both peptide substrates (0–250 mM) using non-linear regression analysis in the GraphPad Prism software. Velocity (V) was calculated from the change in fluorescence/min at the linear phase of the reaction and the Relative Fluorescence Units (RFU) was transformed to molar concentrations by a standard curve for 7-amido-4-methylcoumarin (AMC, Sigma Aldrich).

**PSA activity on protein substrates.** Recombinant protein substrates semenogelin, fibronectin, nidogen-1, laminin α-4 and galectin-3 (R&D Systems) ($0.5\,\mu M$) were incubated with mature $0.2\,\mu M$ recombinant PSA (Wt, Thr[163] and Ala[195]) at 37 °C for 18 h in TBST buffer and analysed by SDS-PAGE analysis[11]. The experiment was repeated twice. The activation rate of pro-MMP2 ($0.14\,\mu M$) and hydrolysis of IGFBP-3 by the PSA protein variants ($0.07\,\mu M$) was determined by a MMP2 screening assay (Abcam) and an immunofluorometric assay to detect intact and total IGFBP-3, respectively. Three technical replicates per group were included. In total three biological replicates were performed.

## HUVEC angiogenesis assays to analyse the anti-angiogenic potential of PSA
The antiangiogenic activity of the PSA protein variants was assessed by the HUVEC tube formation assay[31]. HUVECs were used for tube formation experiments until passage 8 in endothelial growth cell medium (PromoCell)[65,76]. Briefly, four-chamber cell culture slides were coated with Matrigel™ basement membrane preparation (BD Biosciences) and HUVECs ($1.2 \times 10^5$) were added on top of the Matrigel and incubated with conditioned media (200 μL/well) from the stable transfected PC-3-PSA cell line models which were serum starved prior to performing the angiogenesis assay. HUVECs were grown on Matrigel for 18 h, before live cell images were taken using the EVOS fluorescent microscope (AMG, Mill Creek, USA). Five (2× objective) to 14 (4× objective) live cell images for each cell culture chamber were analysed by Fiji ImageJ 1.50b[77] using Angiogenesis Analyzer macro[78]. The following measurements were included in the analysis of angiogenesis index: number of junctions, master junctions, master segments, sum of the length of the detected master segments, and number of meshes and sum of mesh areas detected in the analysed area. Angiogenesis index, reflecting the extent of tube formation or angiogenic potential of the cells, was defined as the average of all these parameters (in relation to control). The angiogenesis index was in keeping with the visual inspection of the images and with the effect of PSA-B in HUVEC tube formation[31,64]. Similarly, the anti-angiogenic potential observed with conditioned media from PSA overexpressing PC-3 cells was verified by 250 nM of the recombinant PSA protein variants Wt, Thr[163] and Ala[195]. Control wells contained an equal amount of phosphate buffered saline (PBS) in culture medium. At least two technical replicates per group in two biological replicates were performed.

## Analysis of the PSA-ACT complex
To analyse the effect of the rs17632542 *KLK3* SNP in complexing of PSA variants with ACT (predominant PSA inhibitor in serum), $0.1\,\mu M$ recombinant mature PSA proteins (generated in *Pichia pastoris*), Wt, Thr[163] and Ala[195] were incubated with ACT ($0.5\,\mu M$) for 15 mins at RT, denatured at 70 °C for 10 min and samples were analysed by SDS-PAGE followed by silver staining.

## Immunohistochemical analysis of patient tissues
FFPE blocks from prostate tumours ($n = 23$) were obtained from the Australian Prostate Cancer Bio-Resource tumour bank. These patients were genotyped for the rs17632542 SNP in our Illumina iSelect genotyping array (iCOGS). A detailed summary on the genotype, age at diagnosis, family history, Gleason Grades, Gleason Score and PSA levels at diagnosis were obtained. Immunohistochemical (IHC) staining was performed using FFPE sections (4 μm) incubated with anti-PSA antibody (1:5000) (Dako; Catalogue number #A0562) overnight at 4 °C followed by incubation with anti-rabbit goat DAB-polymer-linked secondary antibody-based detection (Dako) according to the manufacturer's instructions. Images were acquired using an Olympus VS120 Brightfield slide scanner. All IHC samples were assessed by two independent researchers (a pathologist and an IHC expert) blinded to subject outcomes and sample origin. Each slide was scored for the percentage of PSA positive cells (0% positive cells=0; 1–25% positive cells for 1; 26–50% positive cells for 2; 51–75% positive cells for 3 and >76% positive cells for 4) and staining intensity (no staining = 0; slight staining = 1; moderate staining = 2; strong staining = 3). Scores for both intensity and percentage of positive cells were summed for an overall staining score. The difference in the levels of expression of PSA depending on the patient's allele ([T] vs [C]) for the rs17632542 SNP were then analysed.

## Study populations and genotyping
All cohorts were approved by each study' institutional review board (IRB) and informed consent was obtained from each participant in accordance with principles of the the Declaration of Helsinki. Data from the Australian Prostate Cancer BioResource were approved by the Queensland University of Technology review boards (QUT#1000001165). All studies included in the PRACTICAL Consortium were approved by the respective institution review boards[16–18]. Written informed consent was obtained from each participant and did not receive any compensation other than reimbursement of travel expenses for study appointments. Data from two large population-based studies, the Malmö Diet and Cancer (MDC) and the Västerbotten Intervention Project (VIP) were approved by local institutional review boards (Research Ethics Board at Umeå University, number 2009-1436-31, for VIP and the Research Ethics Board at Lund University, numbers 617/2005 and LU 425-02, for MDC) and written informed consent was obtained from each participant in accordance with the principles of the Declaration of Helsinki and did not receive any compensation.

The rs17632542 SNP was genotyped on the Illumina OncoArray SNP-chip[20] in 81,942 men, which included 49,941 PCa cases and 32,001 disease-free controls[16]. Only samples that were genotypically identified as male (XY) were included in the analysis. The OncoArray Consortium, a large collaborative effort to gain new insight into the genetic architecture underlying breast, ovarian, prostate, colorectal and lung cancers, developed a custom high-density genotyping array, the "OncoArray", that included 310,000 SNPs for meta-analyses and fine-mapping for the above five cancers. Further, 80,000 PCa specific genetic markers derived from previous multi-ethnic meta-analysis[79] (including ancestral populations of Europeans, African Americans, Japanese, and Latin Americans), fine-mapping of known PCa loci, and

candidate nominations were included on the OncoArray. Briefly, 42 studies provided core data on disease status, age at diagnosis (observation or questionnaire for controls), family history, and clinical factors for cases (*e.g.* PSA at diagnosis, Gleason score, etc.) for 49,941 PCa cases and 32,001 disease-free controls. Previous GWAS contributed an additional 32,255 PCa cases and 33,202 disease-free controls of European ancestry for the overall meta-analysis[79]. For survival analysis, 37,316 cases with follow-up on cause-specific death were included. Of these, 4629 died of PCa, 3456 died of other causes. Cases by rs17632542 carrier status were TT = 33,281, CT = 3909 and CC = 126.

Demographic and clinical information on the above study participants including age at diagnosis, Gleason score, stage of disease, PSA levels and cause of death were obtained through in-person interviews or medical or death records. Low risk disease was defined as Gleason score ≤ 6, PSA < 10; intermediate risk as Gleason Score = 7 or PSA = 10–20; and high-risk aggressive disease was defined as Gleason score ≥ 8 or PSA > 20. Genotypes were called using Illumina's proprietary GenCall algorithm. Serum tPSA and f/t PSA analysis were reported for 969 PCa patients (CC (2) and CT (97) genotypes compared to TT (870)).

VIP is an ongoing population-based cohort study initiated in 1986 for 43,692 men with more than 20 years of follow-up and includes residents of Västerbotten County, Sweden. A nested case-control design with three controls matched to each index case were available which included 1743 men with a PCa diagnosis. Of these, there were 126 patients with metastatic PCa during follow-up who subsequently died from PCa[80]. Additional PCa cases (*n* = 1223) were available through the Malmö Diet and Cancer (MDC) cohort, a large prospective, population-based study with more than 20-years of follow-up[80]. In this cohort, 1053 cases with available mortality information were used for survival analysis[81]. The genotype data for the rs17632542 SNP and tPSA and fPSA levels for the MDC and VIP cohorts was available through previous GWAS[22,33].

Control serum samples with tPSA and f/t PSA analysis were available from the MDC (*n* = 2,458) and the VIP (*n* = 4810) cohorts. Serum f/t PSA values have already been reported for these two cohorts (both cases and controls)[33]. Genotype for these men for the rs17632542 SNP were determined through Agena Bioscience MassARRAY matrix-assisted laser desorption/ionization (MALDI-TOF).

### Statistical analyses

Association between the rs17632542 SNP and PCa risk was analysed using the per-allele trend test, adjusted for study relevant covariates using logistic regression and seven principal components derived from analysis of the whole iCOGS and OncoArray dataset. Odds ratios (OR) and 95% confidence intervals (95% CI) were derived using SNPTEST or an in-house C++ program. Tests of homogeneity of the ORs across strata were assessed using a likelihood ratio test. The associations between SNP genotypes and PSA level were assessed using linear regression, after log-transformation of PSA levels to correct for skewness. In a case-only analyses, Cox proportional hazards regression was used to estimate associations of the SNP. To assess the association between the *KLK3* c.536 T > C variant and prognosis after a PCa diagnosis, we used time to event analysis with the primary end point being death from PCa or other causes. Survival time was calculated from the date of diagnosis until the date of death from PCa or all causes other than PCa or, if still alive, the date at last follow-up. Survival analyses were limited to cohorts for which follow-up for cases was at least 90% complete and that have at least 5 PCa deaths. Genotype- [CC vs CT vs TT] or allele- [C vs T] specific analysis was performed to attain sufficient power after adjusting to the low SNP frequency. All regression analyses were performed using SPSS, R and Stata 14[16]. To address the effect of the SNP on f/t PSA levels, all models included study site and principal components as covariates. The associations between SNP genotypes and PSA levels were assessed using linear regression in

R, adjusted for age of the subject at the time of blood draw. The tPSA and fPSA values were log-transformed to limit potential bias because of deviation from normality. All statistical tests were two-sided.

For in-vivo subcutaneous models (*n* = 7/group), two mice in Thr[163] PSA group and one mouse in vector group died due to unrelated bacterial infection and were excluded. Unless otherwise stated, for all other biological or biochemical analyses three independent experiments were conducted with results presented as mean +/− standard deviation, and analysed using a Kruskal-Wallis test, Student T-test, one-way ANOVA or two-way ANOVA with a *p*-value of <0.05 considered statistically significant.

### Reporting summary

Further information on research design is available in the Nature Portfolio Reporting Summary linked to this article.

### Data availability

The OncoArray genotype data and relevant covariate information (ancestry, country, principal components, and so forth)[16] generated have been deposited in dbGaP under accession code phs001391.v1.p1 (https://www.ncbi.nlm.nih.gov/projects/gap/cgi-bin/study.cgi?study_id=phs001391.v1.p1). The previous meta-analysis summary results and genotype data are available in dbGaP under accession code phs001081.v1.p1 (https://www.ncbi.nlm.nih.gov/projects/gap/cgi-bin/study.cgi?study_id=phs001081.v1.p1). These datasets contain individual-level SNP data. The complete meta-analysis summary associations statistics are publicly available at the PRACTICAL website (http://practical.icr.ac.uk/blog/). Data are under restricted access due to IRB restrictions or patient identifiability. Data can be accessed via dbGaP or on application to the consortium. Data would be available for the length of time that is reasonably required to perform the analysis of interest. The data for MDC and VIP studies are not publicly available to maintain compliance with European data protection laws. Anonymised data are available after application to the PIs of these studies. Source data are provided with this paper.

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

## Acknowledgements

We would particularly like to thank all the patients and control men who took part in all the studies involved in this work, as well as all the researchers, clinicians, technicians, and administrative staff who have enabled this work to be carried out, and the collaborators in the PRACTICAL consortium. We acknowledge support from the NIHR to the Biomedical Research Centre at The Institute of Cancer and The Royal Marsden NHS Foundation Trust. We thank (in consortia with other cohorts: NSHDS investigators thank) the Biobank Research Unit at Umeå University, Västerbotten Intervention Programme, the Northern Sweden MONICA study and the County Council of Västerbotten for providing data and samples and acknowledge the contribution from Biobank Sweden, supported by the Swedish Research Council (VR 2017-00650). The Australian Prostate Cancer BioResource funding was obtained from a National Health and Medical Research Council Enabling Grant and an infrastructure grant from the Prostate Cancer Foundation of Australia. The authors acknowledge TRI for providing an excellent research environment and core facilities that enabled this research, particularly Preclinical Imaging, Biological Resources Facility, Histology and Microscopy. The authors acknowledge the Australian Research Council for funding the Vevo 2100/LAZR through a LIEF grant (LE150100067), and the Lions Club of Australia and the Mater Foundation for funding the Skyscan 1272 microCT. The Translational Research Institute is supported by grants from the Australian Government. The authors would like to

thank the Australian Prostate Cancer Research Centre-Queensland for their support. Also, thanks to Dr Sally Stephenson (QUT, Brisbane) for providing the mKO2 vector and Dr. Sunderajhan Sekar and Mr. Sanchit Seth for their contribution and help with spheroid assays. We also thank all current and former members of the Clements and Batra labs for their helpful discussions and insight. This work was supported by project grants from the NHMRC, Cancer Council Queensland and PCFA to J.C. and J.B., J.C. was supported by NHMRC Principal Research Fellowship. J.B. was supported by NHMRC Career Development Fellowship, Advance QLD MCR Research Fellowship (AQIRF175-2019RD2), Education and Research Committee (SERC) grant by Pathology Queensland, DoD Prostate Cancer Idea Development Grant (W81XWH-19-1-0343), PCFA research grant and Centre for Genomics and Personalised Health Collaboration and Innovation grants (2023 and 2024). S.S. was supported by a QUTPRA scholarship, Advance QLD ECR Research Fellowship, PCFA John Mills YI Award, SERC grant by Pathology Queensland and Centre for Genomics and Personalised Health Collaboration and Innovation grants (2023 and 2024). TK was supported by a Movember Revolutionary Team Award. N Bock acknowledges support from the NHMRC, PCFA and Movember through a Peter Doherty ECR Fellowship (APP1091734), John Mills YI Award, and MRT Award, respectively, as well as support via an Advance QLD ECR Research Fellowship (AQIRF066-2019RD2) and from the Max Planck Queensland Centre for the Materials Science of Extracellular Matrices. JP is supported by a QUTPRA scholarship. O.M. was supported by grants from The European Research Council, The Knut and Alice Wallenberg Foundation and the Swedish Research Council. Contract grant sponsor Ana Vega: supported by Spanish Instituto de Salud Carlos III (ISCIII) funding, an initiative of the Spanish Ministry of Economy and Innovation partially supported by European Regional Development FEDER Funds (PI22/00589, PI19/01424; INT20/00071); the ERAPerMed JTC2018 funding (AC18/00117); the Autonomous Government of Galicia (Consolidation and structuring program: IN607B), by the Fundación Mutua Madrileña (call 2018) and by the AECC (PRYES211091VEGA). H.K. is supported by grants from Sigrid Jusélius Foundation and Magnus Ehrnrooth Foundation. R.J.K. was supported by National Cancer Institute [R01 CA175491 and R01 CA244948]. H.L. is supported in part from NIH/NCI by a Cancer Center Support Grant (P30-CA008748), U01-CA199338, R01CA244948, the Swedish Cancer Society (Cancerfonden 23 3074 Pj 01 H), and the Swedish Prostate Cancer Foundation (Prostatacancerförbundet). ABS was supported by NHMRC Investigator Fellowship (APP1177524).

## Author contributions

S.S. performed most of the assays (cell-based assays with PSA overexpression and knockdown, activity assays using recombinant PSA and gene expression studies) and wrote the manuscript. T.K. performed activity analysis and endothelial assays using PC-3 conditioned media. N Bock and JR performed in-vitro bone models. B.T. and K.S. performed in-vivo injections, imaging, and data analysis. P.J. and L.M. helped with gene expression studies, risk analysis and tissue retrieval from the Australian Prostate Cancer BioResource. A.F. generated PSA knockdown models. C.S. performed recombinant PSA expression along with S.S.; Y.D. performed IHC and IHC scoring. S.A. and I.V. performed MSK3 spheroid assays and provided MSK3 cell line. K.B. and J.K. performed FFPE slides marking for tumour and non-malignant regions and IHC scoring. R.N. and R.C.E. performed spheroid image analysis. IMPACT, M.G.D., The Profile Study Steering Committee, J.S. and C.M. provided data for Free/Total PSA analysis. K.M., C.M.T., H.G., N.P., D.A., A.W., J.L.S., S.I.B., L.A.M., S.K., O.C., K.D.S., E.M.G., R.C.T., C.A.H., R.J.M., A.V., F.W., D.E.N., M.K., K.L.P., B.G.N., H.B., E.M.J., M.G., F.C. provided data for survival analysis of PRACTICAL consortium. H.L., O.M. and A.D. contributed GWAS-data for MDC Cohort; and H.L., P.S., G.H., C.H., R.J., E.T. and A.C.R. for SNP-data for VIP Cohort. W.L. and R.J.K. provided the overall survival and metastasis-free survival graphs for the MDC and VIP cohorts. N.Brown, G.D. and B.S. contributed to mice PSA levels and histology analysis for mice H&E slides. T.D., M.B., Z.K. and R.E. performed SNP analysis. A.B.S. has mentored and provided feedback on some aspects of genetics and manuscript editing. H.K. and U.S. performed endothelial cell assays with recombinant PSA and IGFP3 assay. H.L. and R.J.K. has provided the patients and disease-free controls genotype and PSA data for MDC and VIP cohorts. PRACTICAL consortium provided SNP data. The Australian Prostate Cancer BioResource provided DNA samples for genotyping and tissue samples. J.B. and J.C. conceived the project and provided ongoing oversight of the project. All authors contributed to writing, critical reviewing and/or editing the manuscript.

## Competing interests

The patents mentioned herewith are not directly related to this study. H.L. is named on a patent for a statistical method to detect prostate cancer. The patent for the statistical model has been licensed and commercialised as the 4Kscore by OPKO Diagnostics. H.L. receives royalties from sales of this test and owns stock in OPKO. H.L. serves on SAB for Fujirebio Diagnostics. R.E. has the following conflicts of interest to declare: Honoraria from GU-ASCO, Janssen, University of Chicago, Dana Farber Cancer Institute USA as a speaker. Educational honorarium from Bayer and Ipsen, member of external expert committee to Astra Zeneca UK and Member of Active Surveillance Movember Committee. She is a member of the SAB of Our Future Health. She undertakes private practice as a sole trader at The Royal Marsden NHS Foundation Trust and 90 Sloane Street SW1X 9PQ and 280 Kings Road SW3 4NX, London, UK. All the other authors declare no conflict of interest.

## Additional information

# Article

Srilakshmi Srinivasan[1,2,3], Thomas Kryza[4,173], Nathalie Bock[1,2,173], Brian W. C. Tse [5], Kamil A. Sokolowski[5], Panchadsaram Janaththani[1,2,171], Achala Fernando[1,2,3], Leire Moya[1,2], Carson Stephens[1,2], Ying Dong[1,2], Joan Röhl [1,172], Saeid Alinezhad[1,2], Ian Vela[1,6], Joanna L. Perry-Keene[7], Katie Buzacott[7], Robert Nica[8], The IMPACT Study, Manuela Gago-Dominguez[9], The PROFILE Study Steering Committee, Johanna Schleutker [10,11], Christiane Maier[12], Kenneth Muir [13], Catherine M. Tangen[14], Henrik Gronberg [15], Nora Pashayan [16,17], Demetrius Albanes [18], Alicja Wolk [19], Janet L. Stanford[20,21], Sonja I. Berndt[18], Lorelei A. Mucci[22], Stella Koutros[18], Olivier Cussenot[23,24], Karina Dalsgaard Sorensen [25,26], Eli Marie Grindedal[27], Ruth C. Travis[28], Christopher A. Haiman[29], Robert J. MacInnis [30,31], Ana Vega [32,33,34], Fredrik Wiklund[15], David E. Neal[35,36,37], Manolis Kogevinas[38,39,40,41], Kathryn L. Penney[42], Børge G. Nordestgaard [43,44], Hermann Brenner [45,46,47], Esther M. John[48], Marija Gamulin [49], Frank Claessens [50], Olle Melander[51], Anders Dahlin[51], Pär Stattin[19], Göran Hallmans[52], Christel Häggström [53], Robert Johansson[53], Elin Thysell[54], Ann-Charlotte Rönn [55], Weiqiang Li[56], Nigel Brown[57], Goce Dimeski[57], Benjamin Shepherd[58], Tokhir Dadaev [59], Mark N. Brook [59], Amanda B. Spurdle [60], Ulf-Håkan Stenman[61], Hannu Koistinen [61,62], Zsofia Kote-Jarai [59,63], Robert J. Klein [56], Hans Lilja [64,65,66], Rupert C. Ecker [1,2,8], Rosalind Eeles [59,63], The Practical Consortium, The Australian Prostate Cancer BioResource, Judith Clements [1,2,174] & Jyotsna Batra [1,2,3,174] ✉

[1]School of Biomedical Sciences, Faculty of Health, Queensland University of Technology, Brisbane, Queensland (QLD), Australia. [2]Translational Research Institute, Queensland University of Technology, Woolloongabba, Brisbane, QLD, Australia. [3]Centre for Genomic and Personalised Health, Queensland University of Technology, Brisbane, QLD, Australia. [4]Mater Research Institute - The University of Queensland, Translational Research Institute, Woolloongabba, Brisbane, QLD, Australia. [5]Preclinical Imaging Facility, Translational Research Institute, Woolloongabba, Brisbane, QLD, Australia. [6]Department of Urology, Princess Alexandra Hospital, Brisbane, Woolloongabba, Brisbane, QLD, Australia. [7]Pathology Queensland, Sunshine Coast University Hospital Laboratory, Birtinya, Sunshine Coast, QLD, Australia. [8]TissueGnostics GmbH, Vienna, Austria. [9]Health Research Institute of Santiago de Compostela (IDIS), Galicia Public Foundation IDIS, SERGAS, Cancer Genetics and Epidemiology Group, Genomic Medicine Group, Santiago de Compostela, Spain. [10]Institute of Biomedicine, Kiinamyllynkatu 10, FI-20014 University of Turku, Turku, Finland. [11]Department of Medical Genetics, Genomics, Laboratory Division, Turku University Hospital, PO Box 52, 20521 Turku, Finland. [12]Humangenetik Tuebingen, Paul-Ehrlich-Str 23, D-72076 Tuebingen, Germany. [13]Division of Population Health, Health Services Research and Primary Care, University of Manchester, Manchester M13 9PL, UK. [14]SWOG Statistical Center, Fred Hutchinson Cancer Research Center, Seattle, WA, USA. [15]Department of Medical Epidemiology and Biostatistics, Karolinska Institute, SE-171 77 Stockholm, Sweden. [16]Department of Applied Health Research, University College London, London WC1E 7HB, UK. [17]Centre for Cancer Genetic Epidemiology, Department of Public Health and Primary Care, University of Cambridge, Strangeways Laboratory, Worts Causeway, Cambridge CB1 8RN, UK. [18]Division of Cancer Epidemiology and Genetics, National Cancer Institute, NIH, Bethesda, MD 20892, USA. [19]Institute of Environmental Medicine, Karolinska Institutet, 177 77 Stockholm, Sweden. [20]Division of Public Health Sciences, Fred Hutchinson Cancer Research Center, Seattle, WA 98109-1024, USA. [21]Department of Epidemiology, School of Public Health, University of Washington, Seattle, WA 98195, USA. [22]Department of Epidemiology, Harvard T. H. Chan School of Public Health, Boston, MA 02115, USA. [23]CeRePP, Tenon Hospital, F-75020 Paris, France. [24]Sorbonne Universite, GRC n°5, AP-HP, Tenon Hospital, 4 rue de la Chine, F-75020 Paris, France. [25]Department of Molecular Medicine, Aarhus University Hospital, Palle Juul-Jensen Boulevard 99, 8200 Aarhus N, Denmark. [26]Department of Molecular Medicine (MOMA), Aarhus University Hospital, DK-8200 Aarhus N., Denmark. [27]Department of Medical Genetics, Oslo University Hospital, 0424 Oslo, Norway. [28]Cancer Epidemiology Unit, Nuffield Department of Population Health, University of Oxford, Oxford OX3 7LF, UK. [29]Center for Genetic Epidemiology, Department of Preventive Medicine, Keck School of Medicine, University of Southern California/Norris Comprehensive Cancer Center, Los Angeles, CA 90015, USA. [30]Centre for Epidemiology and Biostatistics, Melbourne School of Population and Global Health, The University of Melbourne, Grattan Street, Parkville, VIC 3010, Australia. [31]Cancer Epidemiology Division, Cancer Council Victoria, 615 St Kilda Road, Melbourne, VIC 3004, Australia. [32]Fundación Pública Galega Medicina Xenómica, Santiago de Compostela 15706, Spain. [33]Instituto de Investigación Sanitaria de Santiago de Compostela, Santiago de Compostela 15706, Spain. [34]Centro de Investigación en Red de Enfermedades Raras (CIBERER), Santiago de Compostela, Spain. [35]Nuffield Department of Surgical Sciences, University of Oxford, Oxford, England. [36]University of Cambridge, Department of Oncology, Box 279, Addenbrooke's Hospital, Hills Road, Cambridge CB2 0QQ, UK. [37]Cancer Research UK, Cambridge Research Institute, Li Ka Shing Centre, Cambridge CB2 0RE, UK. [38]ISGlobal, Barcelona Institute for Global Health, Barcelona, Spain. [39]IMIM (Hospital del Mar Research Institute), Barcelona, Spain. [40]Universitat Pompeu Fabra (UPF), Barcelona, Spain. [41]CIBER Epidemiología y Salud Pública (CIBERESP), 28029 Madrid, Spain. [42]Channing Division of Network Medicine, Department of Medicine, Brigham and Women's Hospital/Harvard Medical School, Boston, MA 02115, USA. [43]Faculty of Health and Medical Sciences, University of Copenhagen, 2200 Copenhagen, Denmark. [44]Department of Clinical Biochemistry, Herlev and Gentofte Hospital, Copenhagen University Hospital, Herlev, 2200 Copenhagen, Denmark. [45]Division of Clinical Epidemiology and Aging Research, German Cancer Research Center (DKFZ), D-69120 Heidelberg, Germany. [46]German Cancer Consortium (DKTK), German Cancer Research Center (DKFZ), D-69120 Heidelberg, Germany. [47]Division of Preventive Oncology, German Cancer Research Center (DKFZ), Im Neuenheimer Feld 460, 69120 Heidelberg, Germany. [48]Departments of Epidemiology & Population Health and of Medicine, Division of Oncology, Stanford Cancer Institute, Stanford University School of Medicine, Stanford, CA 94304, USA. [49]School of Medicine, University of Zagreb, Salata 3, 10 000 Zagreb, Croatia. [50]Molecular Endocrinology Laboratory, Department of Cellular and Molecular Medicine, KU Leuven BE-3000, Belgium. [51]Department of Clinical Sciences Malmö, Lund University, Malmö, Sweden. [52]Department of Public Health and Clinical Medicine, Nutritional Research, Umeå University, Umeå, Sweden. [53]The Biobank Research Unit, Umeå University, Umeå, Sweden. [54]Department of Medical Biosciences, Pathology, Umeå University, Umeå, Sweden. [55]Translational Analysis in Molecular Medicine, Karolinska University Hospital, Huddinge, Sweden. [56]Icahn Institute for Data Science and Genome Technology, Department of Genetics and Genomic Sciences, Icahn School of Medicine at Mount Sinai, New York, NY, USA. [57]Department of Chemical Pathology, Pathology Queensland, Princess Alexandra Hospital, Woolloongabba, Brisbane, QLD, Australia. [58]Department of Anatomical Pathology, Pathology Queensland, Princess Alexandra Hospital, Woolloongabba, Brisbane, QLD, Australia. [59]The Institute of Cancer Research, London SM2 5NG, UK. [60]Molecular Cancer Epidemiology Laboratory, QIMR Berghofer Medical Research Institute, Herston, Brisbane, QLD, Australia. [61]Department of Clinical Chemistry and Haematology, University of Helsinki, Helsinki, Finland. [62]HUS Diagnostic Center, Helsinki University Hospital, Helsinki, Finland. [63]Royal Marsden NHS Foundation Trust, London, UK. [64]Department of Pathology and Laboratory Medicine, Memorial Sloan Kettering Cancer

Center, New York, NY, USA. [65]Department of Surgery (Urology Service) and Medicine (Genitourinary Oncology), Memorial Sloan Kettering Cancer Center, New York, NY, USA. [66]Department of Translational Medicine, Lund University, Malmö, Sweden. [171]Present address: Florey Institute of Neuroscience and Mental Health, University of Melbourne, Parkville, Melbourne, VIC, Australia. [172]Present address: Faculty of Health Sciences and Medicine, Bond University, 14 University Drive, Robina, QLD 4226, Australia. [173]These authors contributed equally: Thomas Kryza, Nathalie Bock. [174]These authors jointly supervised this work: Judith Clements, Jyotsna Batra. ✉e-mail: jyotsna.batra@qut.edu.au

## The IMPACT Study

Rosalind Eeles [59,63], Elizabeth Bancroft[63], Elizabeth Page[59], Mark N. Brook [59], Zsofia Kote-Jarai [59,63], Audrey Ardern-Jones[63], Chris Bangma[67], Elena Castro[68], David Dearnaley[59,63], Diana Eccles[69], Gareth Evans[70], Jorunn Eyfjord[71], Alison Falconer[72], Christopher Foster[73,74], Henrik Gronberg [15], Freddie C. Hamdy[35,75], Óskar Þór Jóhannsson[76], Vincent Khoo[63], Hans Lilja [64,65,66], Geoffrey Lindeman[77], Jan Lubinski[78], Lovise Maehle[27], Alan Millner[63], Christos Mikropoulos[79], Anita Mitra[80], Clare Moynihan[59], Judith Offman[81,82], Gad Rennert[83], Lucy Side[84], Mohnish Suri[85] & Penny Wilson[86]

[67]Erasmus University Medical Center, Rotterdam, The Netherlands. [68]Spanish National Cancer Research Center, Madrid, Spain. [69]University of Southampton, Southampton, UK. [70]St Mary's Hospital, Manchester, UK. [71]University of Iceland, Reykjavik, Iceland. [72]Imperial College Healthcare NHS Trust, London, UK. [73]HCA Pathology Laboratories, London, UK. [74]The University of Liverpool, Liverpool, UK. [75]Faculty of Medical Science, University of Oxford, John Radcliffe Hospital, Oxford, UK. [76]Landspitali - National University Hospital of Iceland, Reykjavik, Iceland. [77]The Walter and Eliza Hall Institute of Medical Research, Parkville, VIC, Australia. [78]International Hereditary Cancer Center, Szczecin, Poland. [79]Medway Hospital, Kent, UK. [80]University College London Hospitals NHS Foundation Trust, London, UK. [81]Queen Mary University of London, London, UK. [82]Guy's Hospital, London, UK. [83]CHS National Cancer Control Center, Carmel Medical Center, Haifa, UK. [84]Wessex Clinical Genetics Service, Southampton, UK. [85]Nottingham City Hospital, Nottingham, UK. [86]Innovate, Nottingham, UK.

## The PROFILE Study Steering Committee

Rosalind Eeles [59,63], David E. Neal[35,36,37], Freddie C. Hamdy[35,75], Pardeep Kumar[63], Zsofia Kote-Jarai [59,63], Judith Offman[81,82], Antonis Antoniou[87], Jana McHugh[59], Holly Ni Raghallaigh[59], Rose Hall[59], Elizabeth Bancroft[63], Natalie Taylor[63], Sarah Thomas[63], Kathryn Myhill[63], Matthew Hogben[63], Eva McGrowder[59], Elizabeth Page[59], Mark N. Brook [59], Diana Keating[59], Denzil James[59], Joe Merson[59], Syed Hussain[59], Angela Wood[59], Nening Dennis[59], Audrey Ardern-Jones[63], Paul Ardern-Jones[88], Nick van As[63], Elena Castro[68], David Dearnaley[59,63], Christopher Foster[73,74], Steve Hazell[63], Vincent Khoo[63], Sarah Lewis[89], Hans Lilja [64,65,66], Clare Moynihan[59], Paul Pharoah[90], Jack Schalken[91], Aslam Sohaib[63], Nandita de Souza[59], Paul Cathcart[80], Frank Chingewundoh[92], Mathew Perry[93], Jeff Bamber[59], Nora Pashayan [16,17], Manolis Kogevinas[38,39,40,41], Alexander Dias[59], Christos Mikropolis[59], Sibel Saya[59], Antony Chamberlain[59], Anne-Marie Borges Da Silva[59], Lucia D'Mello[63], Sue Moss[81], Jane Melia[87], Netty Kinsella[63], Justyna Sobczak[63], Naami Mcaddy[63], David Nicol[63], Chris Ogden[63], Declan Cahill[63], Alan Thompson[63], Christopher Woodhouse[63], Vincent J. Gnanapragasam[94], Colin Cooper[95] & Jeremy Clark[95]

[87]University of Cambridge, Cambridge, UK. [88]Patient Representative, Cambridge, UK. [89]University of Bristol, Bristol, UK. [90]Department of Computational Biomedicine, Cedars-Sinai Medical Center, West Hollywood, USA. [91]Radboud University, Nijmegen, Netherlands. [92]Barts Health NHS Trust, London, UK. [93]St George's University Hospitals NHS Foundation Trust, London, UK. [94]Cambridge University Hospitals NHS Foundation Trust, Cambridge, UK. [95]University of East Anglia, Norwich, UK.

## The Practical Consortium

Fredrick R. Schumacher[96,97], Sara Benlloch[59,98], Ali Amin Al Olama[98,99], Stephen Chanock[18], Ying Wang[100], Stephanie J. Weinstein[18], Catharine M. L. West[101], Géraldine Cancel-Tassin[23,24], Freddie C. Hamdy[35,75], Jenny L. Donovan[102], Robert J. Hamilton[103,104], Sue Ann Ingles[105], Barry S. Rosenstein[106], Yong-Jie Lu[107], Graham G. Giles[30,31,108], Adam S. Kibel[109], Jong Y. Park[110], Cezary Cybulski[111], Sune F. Nielsen[43,44], Jeri Kim[112], Manuel R. Teixeira[113,114,115], Susan L. Neuhausen[116], Kim De Ruyck[117], Azad Razack[118], Lisa F. Newcomb[20,119], Davor Lessel[120], Radka Kaneva[121], Nawaid Usmani[122,123], Paul A. Townsend[124,125], Jose Esteban Castelao[126], Ron H. N. van Shaik[127], Florence Menegaux[128], Kay-Tee Khaw[129], Lisa Cannon-Albright[130,131], Hardev Pandha[125], Stephen N. Thibodeau[132], Peter Kraft[133], William J. Blot[134,135], Artitaya Lophatananon[13], Phyllis J. Goodman[14], Ian M. Thompson Jr.[136], Tobias Nordström[15,137], Alison M. Dunning[17], Teuvo L. J. Tammela[138,139], Anssi Auvinen[140], Niclas Håkansson[141], Gerald L. Andriole[142], Robert N. Hoover[18], Mitchell J. Machiela[18], Edward Giovannucci[143], Laura E. Beane Freeman[18], Michael Borre[144,145], Lovise Maehle[27], Tim J. Key[28], Loic Le Marchand[146], Xin Sheng[29], Melissa C. Southey[31,108,147], Roger L. Milne[30,31,108], Antonio Gómez-Caamaño[148], Laura Fachal[32,33,34,98], Martin Eklund[15], Trinidad Dierssen-Sotos[41,149], Gemma Castaño-Vinyals[38,39,40,41], Antonio Alcaraz[150], Sara Lindström[151], Meir Stampfer[42], Stig E. Bojesen[43,44],

Hein V. Stroomberg[152], Andreas Røder[152], Xin Gao[45], Bernd Holleczek[153], Ben Schöttker[45], Josef Hoegel[154], Thomas Schnoeller[155], Tomislav Kulis[156], Steven Joniau[157], Maria Elena Martinez[158] & Markus Aly[15,159,160]

[96]Seidman Cancer Center, University Hospitals, Cleveland, OH 44106, USA. [97]Department of Population and Quantitative Health Sciences, Case Western Reserve University, Cleveland, OH 44106-7219, USA. [98]Centre for Cancer Genetic Epidemiology, Department of Public Health and Primary Care, University of Cambridge, Strangeways Research Laboratory, Cambridge CB2 0SR, UK. [99]University of Cambridge, Department of Clinical Neurosciences, Stroke Research Group, R3, Box 83, Cambridge Biomedical Campus, Cambridge CB2 0QQ, UK. [100]Department of Population Science, American Cancer Society, 250 Williams Street, Atlanta, GA 30303, USA. [101]Division of Cancer Sciences, University of Manchester, Manchester Academic Health Science Centre, Radiotherapy Related Research, The Christie Hospital NHS Foundation Trust, Manchester M13 9PL, UK. [102]Population Health Sciences, Bristol Medical School, University of Bristol, Bristol BS8 2PS, UK. [103]Dept. of Surgical Oncology, Princess Margaret Cancer Centre, Toronto, ON M5G 2M9, Canada. [104]Dept. of Surgery (Urology), University of Toronto, Toronto, Canada. [105]Department of Preventive Medicine, Keck School of Medicine, University of Southern California/Norris Comprehensive Cancer Center, Los Angeles, CA 90015, USA. [106]Department of Radiation Oncology, Box 1236, Icahn School of Medicine at Mount Sinai, One Gustave L. Levy Place, New York, NY 10029, USA. [107]Centre for Cancer Biomarker and Biotherapeutics, Barts Cancer Institute, Queen Mary University of London, John Vane Science Centre, Charterhouse Square, London EC1M 6BQ, UK. [108]Precision Medicine, School of Clinical Sciences at Monash Health, Monash University, Clayton, VIC 3168, Australia. [109]Division of Urologic Surgery, Brigham and Womens Hospital, 75 Francis Street, Boston, MA 02115, USA. [110]Department of Cancer Epidemiology, Moffitt Cancer Center, 12902 Magnolia Drive, Tampa, FL 33612, USA. [111]International Hereditary Cancer Center, Department of Genetics and Pathology, Pomeranian Medical University, 70-115 Szczecin, Poland. [112]The University of Texas M. D. Anderson Cancer Center, Department of Genitourinary Medical Oncology, 1515 Holcombe Blvd., Houston, TX 77030, USA. [113]Department of Laboratory Genetics, Portuguese Oncology Institute of Porto (IPO Porto) / Porto Comprehensive Cancer Center, Porto, Portugal. [114]Cancer Genetics Group, IPO Porto Research Center (CI-IPOP) / RISE@CI-IPOP (Health Research Network), Portuguese Oncology Institute of Porto (IPO Porto) / Porto Comprehensive Cancer Center, Porto, Portugal. [115]School of Medicine and Biomedical Sciences (ICBAS), University of Porto, Porto, Portugal. [116]Department of Population Sciences, Beckman Research Institute of the City of Hope, 1500 East Duarte Road, Duarte, CA 91010, USA. [117]Ghent University, Faculty of Medicine and Health Sciences, Basic Medical Sciences, Proeftuinstraat 86, B-9000 Gent, Belgium. [118]Department of Surgery, Faculty of Medicine, University of Malaya, 50603 Kuala Lumpur, Malaysia. [119]Department of Urology, University of Washington, 1959 NE Pacific Street, Box 356510 Seattle, WA 98195, USA. [120]Institute of Human Genetics, University Medical Center Hamburg-Eppendorf, D-20246 Hamburg, Germany. [121]Molecular Medicine Center, Department of Medical Chemistry and Biochemistry, Medical University of Sofia, Sofia, 2 Zdrave Str., 1431 Sofia, Bulgaria. [122]Department of Oncology, Cross Cancer Institute, University of Alberta, 11560 University Avenue, Edmonton, AB T6G 1Z2, Canada. [123]Division of Radiation Oncology, Cross Cancer Institute, 11560 University Avenue, Edmonton, AB T6G 1Z2, Canada. [124]Division of Cancer Sciences, Manchester Cancer Research Centre, Faculty of Biology, Medicine and Health, Manchester Academic Health Science Centre, NIHR Manchester Biomedical Research Centre, Health Innovation Manchester, Univeristy of Manchester, M13 9WL Manchester, UK. [125]The University of Surrey, Guildford, Surrey GU2 7XH, UK. [126]Genetic Oncology Unit, CHUVI Hospital, Complexo Hospitalario Universitario de Vigo, Instituto de Investigación Biomédica Galicia Sur (IISGS), 36204 Vigo (Pontevedra), Spain. [127]Department of Clinical Chemistry, Erasmus University Medical Center, Rotterdam, the Netherlands. [128]"Exposome and Heredity", CESP (UMR 1018), Faculté de Médecine, Université Paris-Saclay, Inserm, Gustave Roussy, Villejuif, France. [129]Clinical Gerontology Unit, University of Cambridge, Cambridge CB2 2QQ, UK. [130]Division of Epidemiology, Department of Internal Medicine, University of Utah School of Medicine, Salt Lake City, UT 84132, USA. [131]George E. Wahlen Department of Veterans Affairs Medical Center, Salt Lake City, UT 84148, USA. [132]Department of Laboratory Medicine and Pathology, Mayo Clinic, Rochester, MN 55905, USA. [133]Program in Genetic Epidemiology and Statistical Genetics, Department of Epidemiology, Harvard School of Public Health, Boston, MA, USA. [134]Division of Epidemiology, Department of Medicine, Vanderbilt University Medical Center, 2525 West End Avenue, Suite 800, Nashville, TN 37232, USA. [135]International Epidemiology Institute, Rockville, MD 20850, USA. [136]CHRISTUS Santa Rosa Hospital – Medical Center, San Antonio, TX, USA. [137]Department of Clinical Sciences at Danderyds Hospital, Karolinska Institute, Stockholm, Sweden. [138]Department of Urology, Tampere University Hospital, FI-33521 Tampere, Finland. [139]Faculty of Medicine and Health Technology, Tampere University, FI-33100 Tampere, Finland. [140]Unit of Health Sciences, Faculty of Social Sciences, Tampere University, Tampere, Finland. [141]Unit of Cardiovascular and Nutritional Epidemiology, Institute of Environmental Medicine, Karolinska Institutet, SE-171 77 Stockholm, Sweden. [142]The Washington University School of Medicine, 660 S. Euclid Avenue, Campus Box 8242, St. Louis, MO 63110, USA. [143]Department of Epidemiology, Harvard School of Public Health, Boston, MA 02115, USA. [144]Department of Urology, Aarhus University Hospital, Palle Juul-Jensen Boulevard 99, 8200 Aarhus N, Denmark. [145]Department of Clinical Medicine, Aarhus University, DK-8200 Aarhus N, Denmark. [146]Epidemiology Program, University of Hawaii Cancer Center, Honolulu, HI 96813, USA. [147]Department of Clinical Pathology, The Melbourne Medical School, The University of Melbourne, Melbourne, VIC 3010, Australia. [148]Department of Radiation Oncology, Complexo Hospitalario Universitario de Santiago, SERGAS, Santiago de Compostela 15706, Spain. [149]University of Cantabria-IDIVAL, Santander, Spain. [150]Department and Laboratory of Urology. Hospital Clínic. Institut d'Investigacions Biomèdiques August Pi i Sunyer (IDIBAPS), Universitat de Barcelona. Spain, C/Villarroel 170, 08036 Barcelona, Spain. [151]Department of Epidemiology, Health Sciences Building, University of Washington, Washington, USA. [152]Copenhagen Prostate Cancer Center, Department of Urology, Rigshospitalet, Copenhagen University Hospital, DK-2730Herlev, Copenhagen, Denmark. [153]Saarland Cancer Registry, 66119 Saarbrücken, Germany. [154]Institute for Human Genetics, University Hospital Ulm, Albert-Einstein-Allee 11, 89081 Ulm, Germany. [155]Department of Urology, University Hospital Ulm, Ulm, Germany. [156]Department of Urology, University Hospital Center Zagreb, University of Zagreb School of Medicine, Zagreb, Croatia. [157]Department of Urology, University Hospitals Leuven, Herestraat 49, Box 7003 41, BE-3000 Leuven, Belgium. [158]Moores Cancer Center, Department of Family Medicine and Public Health, University of California San Diego, La Jolla, CA 92093-0012, USA. [159]Department of Molecular Medicine and Surgery, Karolinska Institute, stockholm, Sweden. [160]Department of Urology, Karolinska University Hospital, Solna, 171 76 Stockholm, Sweden.

## The Australian Prostate Cancer BioResource

Wayne Tilley[161], Gail P. Risbridger[162,163,164], Judith Clements [1,2,174], Lisa Horvath[165,166], Renea Taylor[163,164,167], Lisa Butler[168,169], Anne-Maree Haynes[166,170], Melissa Papargiris[162], Ian Vela[1,2,6], Leire Moya[1,2] & Jyotsna Batra [1,2,3,174]✉

[161]Dame Roma Mitchell Cancer Research Centre, University of Adelaide, Adelaide, SA, Australia. [162]Department of Anatomy and Developmental Biology, Biomedicine Discovery Institute Cancer Program, Monash University, Melbourne, VIC, Australia. [163]Prostate Cancer Research Program, Cancer Research Division, Peter MacCallum Cancer Centre, Melbourne Victoria, Australia. [164]Sir Peter MacCallum Department of Oncology, University of Melbourne,

Melbourne, VIC, Australia. [165]Chris O'Brien Lifehouse (COBLH), Camperdown, NSW, Australia. [166]Garvan Institute of Medical Research, Sydney, NSW, Australia. [167]Department of Physiology, Biomedicine Discovery Institute Cancer Program, Monash University, Melbourne, VIC, Australia. [168]Adelaide Medical School and Freemasons Foundation Centre for Men's Health, University of Adelaide, Adelaide, SA 5005, Australia. [169]South Australian Health and Medical Research Institute, Adelaide, SA 5001, Australia. [170]The Kinghorn Cancer Centre (TKCC), Victoria, NSW, Australia.

