## [Peer Review File · Nature Communications]

A PSA SNP associates with cellular function and clinical outcome in men with prostate cancer.Editorial Note: This manuscript has been previously reviewed at another journal. This document only contains reviewer comments and rebuttal letters for versions considered at Nature Communications.

Reviewers' comments:

Reviewer #1 (Remarks to the Author):

The authors have made efforts to refine the manuscript and resolve some initial confusions, which is greatly appreciated. However, it's of concern that no additional experiments have been carried out or new data provided to address key issues pointed out by the three reviewers. Significant points of concern, including the lack of additional validation of KLK3-OE beyond a single AR-negative cell line (PC3), remain unaddressed, hindering the manuscript's scientific rigor and validity.

Reviewer #3 (Remarks to the Author):

The authors have described a very interesting study investigating a germline variant of PSA and its association with prostate cancer outcomes. The authors have demonstrated that overexpression of this PSA variant decreased proliferation of androgen-independent cancer cells but unexpectedly increased metastatic-associated traits in vitro and in vivo. They go on to demonstrate that the I163T variant has reduced peptidase activity, reduced ability to inhibit angiogenesis, yet reduced ability to complex with protease inhibitors. Analysis of patient cohorts confirmed that rs17632542 SNP is associated with reduced risk of prostate cancer. However, analysis of patients who died in these cohorts revealed a novel and unexpected enrichment of this I163T SNP.

Genuinely appreciate the improvements made to the text, as well as the responses to earlier review. To my eyes, the experiments presented appear very similar to the prior version.

Critical Issues, both highlighted previously:

I. One of the fundamental points regarding this manuscript's novelty and impact is its assertion that beyond serving as a correlative biomarker, PSA has substantial biologic impact on the metastasis of prostate cancer cells, as well as related characteristics in vitro. Two of the four main figures are dedicated to this point. This assertion is based on the overexpression of PSA variants in androgen independent cell models i.e. cells that do not express PSA. There is a grave concern that these findings are simply artefactual, and do not reflect the behaviour of cells that actually express PSA variants. It is critical that such experiments be robustly performed in androgen dependent models. If CRISPR editing in the mutations is challenging, one alternative approach is to knockout KLK3 completely, and then rescue it with a (possibly CRISPR-resistant) KLK3 variant. Could also rescue with multiple variants simultaneously to model the CT variant, as it is unclear what is the significance of CT heterozygous phenotype. As the experiments stand, it would be similar to over-expressing PSA in another cancer cell type, and seeing a biologic effect (which would not be surprising, given the enzymatic activity).

II. Concerns about the relevance of the KLK3 experimental investigations then calls into question the underlying reason for the counterintuitive findings that the PSA I163T is associated with decreased prostate cancer incidence, but is also enriched among those who die from prostate cancer. It should be noted that the relationship between prostate cancer metastasis and death is complex: upon diagnosis of metastasis, patients have a median survival of 5 additional years and much longer than other cancers. The development of castration-resistant metastases, or therapy-resistant metastases, contributes substantially to premature death. All the more reason to perform experimentation in an androgen-dependent model. Ascribing earlier mortality due to a non-proliferative, purely metastatic biology, with agnostic effects regarding AR signaling/inhibition, is a hypothesis that needs further experimentation.

There are other reasonable hypotheses to explain the Figure 4 finding. It would seem that if detection occurs at a later time for C-allele patients who do end up developing prostate cancer, that could explain the 'paradoxical' reason for C-allele enrichment among those who die from it.

Reviewer #4 (Remarks to the Author): Expert in prostate cancer organoids, preclinical models, and

biomarkers; replaces Reviewer #2

I was assigned the task to go over the rebuttal in reference to the comments of one of the reviewers. As the Reviewer points out "All experiments are performed on one cell line, i.e. PC3" this cell line is certainly not representative of PCa. However the inclusion of MSK organoids does support the hypothesis.

The authors should uniform nomenclature of the MSK, they are defined as patient derived organoids, organoids and spheroids...also the term spheroid is also used to determine a morphological parameters, which is confusing.

Reviewer's Comments:

Reviewer #1 (Remarks to the Author)

The authors have made efforts to refine the manuscript and resolve some initial confusions, which is greatly appreciated. However, it's of concern that no additional experiments have been carried out or new data provided to address key issues pointed out by the three reviewers. Significant points of concern, including the lack of additional validation of KLK3-OE beyond a single AR-negative cell line (PC3), remain unaddressed, hindering the manuscript's scientific rigor and validity.

Answer: We have now incorporated new experimental data (Figure 1 below) to address critical points raised by the Reviewers regarding the validation of our findings in AR-positive cell lines. Following the suggestion of Reviewer 2, we conducted stable knockdown of KLK3 in LNCaP cells, an AR-positive and high PSA-expressing cell line. Subsequently, established KLK3 knockdown cells, with over 90% knockdown efficiency, were re-transfected with PSA isoforms (Wt, Thr¹⁶³, Ala¹⁹⁵) and vector. Our updated LNCaP expression models consistently confirmed the functional effects observed earlier in the PC-3 and MSK3 models (Figure 1B, 1E, Figure 2E-G, Supplementary Figure 1A).

Figure 1: A) Stable knock-down of PSA in LNCaP cells (>90% knockdown efficiency) and re-transfection with PSA variants (Wt, Thr¹⁶³, Ala¹⁹⁵ and Vector). **(B)** Proliferation and **(C)** migration assays showing higher proliferation of Wt PSA expressing LNCaP-PSA cells consistent with our previous observations in PC-3 and patient-derived organoid MSK3 overexpression models. **(D)** Spheroid assays demonstrate the higher invasive ability of the Thr¹⁶³ PSA expressing LNCaP cells with a higher peripheral area and less spherical inner core.

Additionally, contrary to the impression that may have been conveyed, we did indeed conduct the experiment in an additional AR positive cell line: a patient derived organoid MSK3 cell line (PMID: 25201530 – Please see figure below for the AR expression in MSK-PCa3 cells).

To further confirm the expression of AR in all the cell line models in our study, we have recently carried out the expression analysis of AR using qPCR and observed the PSA to regulate AR expression as below. This figure is now included in Supplementary Figure 1B.

Reviewer #3 (Remarks to the Author)

The authors have described a very interesting study investigating a germline variant of PSA and its association with prostate cancer outcomes. The authors have demonstrated that overexpression of this PSA variant decreased proliferation of androgen-independent cancer cells but unexpectedly increased metastatic-associated traits in vitro and in vivo. They go on to demonstrate that the I163T variant has reduced peptidase activity, reduced ability to inhibit angiogenesis, yet reduced ability to complex with protease inhibitors. Analysis of patient cohorts confirmed that rs17632542 SNP is associated with reduced risk of prostate cancer. However, analysis of patients who died in these cohorts revealed a novel and unexpected enrichment of this I163T SNP.

Genuinely appreciate the improvements made to the text, as well as the responses to earlier review. To my eyes, the experiments presented appear very similar to the prior version.

Thank you for the summary and for appreciating our efforts addressing Reviewers' comments in our previous version. Relevant to the comments made by Reviewer#3, we repeated the cell-based experiments in LNCaP cells as discussed and shown in the Figure 1 above and have incorporated the new data in our revised manuscript.

Critical Issues, both highlighted previously:

I. One of the fundamental points regarding this manuscript's novelty and impact is its assertion that beyond serving as a correlative biomarker, PSA has substantial biologic impact on the metastasis of prostate cancer cells, as well as related characteristics in vitro. Two of the four main figures are dedicated to this point. This assertion is based on the overexpression of PSA variants in androgen independent cell models i.e. cells that do not express PSA. There is a grave concern that these findings are simply artefactual, and do not reflect the behaviour of cells that actually express PSA variants. It is critical that such experiments be robustly performed in androgen dependent models. If CRISPR editing in the mutations is challenging, one alternative approach is to knockout KLK3 completely, and then rescue it with a (possibly CRISPR-resistant) KLK3 variant. Could also rescue with multiple variants simultaneously to model the CT variant, as it is unclear what is the significance of CT heterozygous phenotype. As the experiments stand, it would be similar to over-expressing PSA in another cancer cell type, and seeing a biologic effect (which would not be surprising, given the enzymatic activity).

Answer: We thank the Reviewer for these insightful comments. We have followed the advice and have performed shRNA-mediated knockdown of the KLK3 gene in AR positive LNCaP prostate cancer cells, followed by replenishing with mammalian vectors expressing the Wt and Thr¹⁶³ PSA variants (Figure 1 above). Our updated LNCaP expression models consistently confirmed the functional effects observed earlier in the PC-3 and MSK3 models.

Additionally, the data presented on patients derived organoids i.e. MSK3 cell lines are AR positive cell lines. We apologise not making this clear in our previous submission and have now been clarified.

II. Concerns about the relevance of the KLK3 experimental investigations then calls into question the underlying reason for the counterintuitive findings that the PSA I163T is associated with decreased prostate cancer incidence but is also enriched among those who die from prostate cancer. It should be noted that the relationship between prostate cancer metastasis and death is complex: upon diagnosis of metastasis, patients have a median survival of 5 additional years and much longer than other cancers. The development of castration-resistant metastases, or therapy-resistant metastases, contributes substantially to premature death. All the more reason to perform experimentation in an androgen-dependent model. Ascribing earlier mortality due to a non-proliferative, purely metastatic biology, with agnostic effects regarding AR signaling/inhibition, is a hypothesis that needs further experimentation.

Answer: We thank the Reviewer's perspective-taking comment. As we have now included new data in LNCaP AR positive cells and clarified that MSK3 organoids are AR positive (AR +ve), this might resolve concerns of the Reviewers (Figure 1 above). We have now also included this perspective in our discussion in the revised version (page 12, lines 21-27)

There are other reasonable hypotheses to explain the Figure 4 finding. It would seem that if detection occurs at a later time for C-allele patients who do end up developing prostate cancer, that could explain the 'paradoxical' reason for C-allele enrichment among those who die from it.

We acknowledge the Reviewers' alternative interpretation of the results, which is also supported by our biochemical work showing the effect of the KLK3 SNP on PSA expression. In response to this

feedback, we have incorporated the following text in our revised version, under the discussion section, page 13, lines 35-36: "The high frequency of the SNP in patients with aggressive cancer could also be attributed to their late diagnosis owing to the low PSA levels."

Reviewer #4 (Remarks to the Author)

I was assigned the task to go over the rebuttal in reference to the comments of on of the reviewers. As the Reviewer points out "All experiments are performed on one cell line, i.e. PC3" this cell line is certainly not representative of PCa. However the inclusion of MSK organoids does support the hypothesis.

The authors should uniform nomenclature of the MSK, they are defined as patient derived organoids, organoids and spheroids...also the term spheroid is also used to determine a morphological parameters, which is confusing.

Answer: We thank the Reviewer for pointing this out. We have now corrected the text to avoid confusion (page 6, line 21; page 6 line 30; page 17 line 9; page 18, line 3; page 18, line 40; page 19, line 8; page 20, line 30; page 21, line 9; page 35; page 37.

REVIEWER COMMENTS

Reviewer #1 (Remarks to the Author):

The authors have addressed my major concerns and the manuscript is significantly improved.

Reviewer #3 (Remarks to the Author):

The authors have a genuinely interesting manuscript testing the hypothesis that PSA / KLK-3 mutants affect the biology and clinical outcomes of prostate cancer patients' tumors possessing such mutations. If correct, would have implications for patients and needs to be assessed rigorously. Appreciate the authors for performing re-analyses, then additional experiments, in response to reviewers' two rounds of input.

Now the authors have included a cell line, LNCaP, that endogenously expresses KLK3. It is unclear to me whether LNCaPs express wild-type KLK3 or a mutated KLK3, based on the study by Spans et al, PMID 24587179, since this was not addressed in the rebuttal manuscript. That aside, my reading of the rebuttal experiments, is that knocking down LNCaP KLK3, followed by overexpression of wt KLK3 (10-fold greater mRNA than endogenous) leads to 60% confluence at 72h, whereas overexpression of Thr163 KLK3 (9-fold greater mRNA than endogenous) leads to 40% confluence. There appear to be many missing easy and routine controls: whether KLK3 knockdown alone (vector) has change on proliferation vs LNCaPs (no vector); Western blots to evaluate KLK3 protein levels following shRNA knockdown and rescue attempts with exogenous KLK3 mutants; use of a second KLK3 shRNA hairpin to reduce chance of studying common off-target effects, LNCaP xenografts to evaluate KLK3 mutants (come up rapidly, not onerous to do).

Hard to conclude with confidence that KLK3 mutations lead to altered biology in prostate cancer patients.

REVIEWER COMMENTS

Reviewer #1 (Remarks to the Author):

The authors have addressed my major concerns and the manuscript is significantly improved.

Thank you for appreciating our efforts addressing the Reviewers' comments.

Reviewer #3 (Remarks to the Author):

The authors have a genuinely interesting manuscript testing the hypothesis that PSA / KLK-3 mutants affect the biology and clinical outcomes of prostate cancer patients' tumors possessing such mutations. If correct, would have implications for patients and needs to be assessed rigorously. Appreciate the authors for performing re-analyses, then additional experiments, in response to reviewers' two rounds of input.

We thank the Reviewer for acknowledging our re-analyses and inclusion of the additional data in our revised version.

Now the authors have included a cell line, LNCaP, that endogenously expresses KLK3. It is unclear to me whether LNCaPs express wild-type KLK3 or a mutated KLK3, based on the study by Spans et al, PMID 24587179, since this was not addressed in the rebuttal manuscript.

LNCaPs harbor homozygous TT genotype and express wild-type KLK3 as previously stated in our main text, page 17, line 15. We have obtained the sequence from our in-house RNA-seq data and have also confirmed this through our recent Sanger sequencing (sequencing results shown below with the SNP position highlighted)

Human DNA for prostate specific antigen (PSA)

Sequence ID: X14810.1 Length: 5873 Number of Matches: 1

Range 1: 3681 to 3980 GenBank Graphics

▼ Next Match ▲ Previous Match

Score	Expect	Identities	Gaps	Strand
555 bits(300)	3e-153	300/300(100%)	0/300(0%)	Plus/Plus
Query 1	GGTCATGGACCTGCCACCCAGGAGCCAGCACTGGGGACCACCTGCTACGCCTCAGGCTG			
Sbjct 3681	GGTCATGGACCTGCCACCCAGGAGCCAGCACTGGGGACCACCTGCTACGCCTCAGGCTG			
Query 61	GGGCAGCATTGAACCAGAGGAGTGTACGCCTGGGCCAGATGGTGCAGCCGGGAGCCAGA			
Sbjct 3741	GGGCAGCATTGAACCAGAGGAGTGTACGCCTGGGCCAGATGGTGCAGCCGGGAGCCAGA			
Query 121	TGCCTGGGTCTGAGGGAGGAGGGGACAGGACTCCTGGGTCTGAGGGAGGAGGGCCAAGGA			
Sbjct 3801	TGCCTGGGTCTGAGGGAGGAGGGGACAGGACTCCTGGGTCTGAGGGAGGAGGGCCAAGGA			
Query 181	ACCAGGTGGGGTCCAGCCACAACAGTGTGTTTTGCCTGGCCCGTAGTCTTGACCCCAAAG			
Sbjct 3861	ACCAGGTGGGGTCCAGCCACAACAGTGTGTTTTGCCTGGCCCGTAGTCTTGACCCCAAAG			
Query 241	AAACTTCAGTGTGTGGACCTCCATGTTATTTCCAATGACGTGTGTGCGCAAGTTCACCT			
Sbjct 3921	AAACTTCAGTGTGTGGACCTCCATGTTATTTCCAATGACGTGTGTGCGCAAGTTCACCT			

That aside, my reading of the rebuttal experiments, is that knocking down LNCaP KLK3, followed by overexpression of wt KLK3 (10-fold greater mRNA than endogenous) leads to 60% confluence at 72h, whereas overexpression of Thr163 KLK3 (9-fold greater mRNA than endogenous) leads to 40%

confluence. There appear to be many missing easy and routine controls: whether KLK3 knockdown alone (vector) has change on proliferation vs LNCaPs (no vector).

LNCaP cells transfected with non-target control behaved similarly to the LNCaP cells alone. Endogenous KLK3 (Wt PSA) knockdown in non-transfected LNCaP cells slightly reduced proliferation, although this was not statistically significant compared to the shKLK3+vector control. This modified figure including control LNCaP cells is included in the Supplementary Figure 1C.

Western blots to evaluate KLK3 protein levels following shRNA knockdown and rescue attempts with exogenous KLK3 mutants.

Secreted serum total PSA levels in cell line conditioned media for the generated LNCaP overexpression models were measured by ELISA in line with our previous results which are now also included in Supplementary Table 1.

Cell line model	Protein name	Total PSA ($\mu\text{g/L}$)
MSK3	Wt PSA	56.4
MSK3	Thr ¹⁶³ PSA	48.5
MSK3	Ala ¹⁹⁵ PSA	68.4
MSK3	Vector	0.5
LNCaP	Wt PSA	33.2
LNCaP	Thr ¹⁶³ PSA	27.8
LNCaP	Ala ¹⁹⁵ PSA	54.2
LNCaP	shRNA + vector	1.2
LNCaP	Control	6.0

Use of a second KLK3 shRNA hairpin to reduce chance of studying common off-target effects, LNCaP xenografts to evaluate KLK3 mutants (come up rapidly, not onerous to do).

We have used a 3'-UTR targeting shRNA, to knockdown the endogenous PSA expression in LNCaP cells to undertake subsequent re-transfection with the PSA variants. Our primary comparison is to assess the functionality between the PSA variants, we therefore, established a stable knockdown model using one shRNA, which has significantly reduced KLK3 expression (both mRNA and protein) and did not drastically affect the cellular properties of cells (proliferation graph included above confirms this).

In a previous study, we have shown miR-3162-5p has strong binding affinity to the T allele of *KLK3* rs1058205 miRSNP using reporter assays (PMID: 25691096). In our second study (PMID: 3101891), using miR-3162-5p mimics, we demonstrated that, miR-3162-5p mediated knock-down of *KLK3* gene, reduced protein levels of KLK3 and proliferation of LNCaP cells by additionally targeting other KLKs (KLK2, KLK4) and AR. These two studies highlight the role of KLK3/PSA in the cellular function of PCa cells and validates our observation for lower proliferation and migration observed in the LNCaP-*KLK3* knockdown models. We have now included this in our discussion (page 11, lines 29-35).

REVIEWERS' COMMENTS

Reviewer #3 (Remarks to the Author):

Biochemical activity induced by a germline variation in KLK3 (PSA) associates with cellular function and clinical outcome in prostate cancer.

Srinivasan et al. have described an interesting study investigating a germline I163T variant of PSA and its association with prostate cancer outcomes. In an earlier manuscript version, the authors demonstrated that high overexpression of this PSA variant decreased proliferation in two KLK3-negative lines (PC3, MSK-PCa3) and unexpectedly increased metastatic-associated traits in vitro and in vivo. In subsequent revisions, they have added a KLK3-expressing cell line (LNCaP). They went on to demonstrate in vitro that the I163T variant has reduced peptidase activity, reduced ability to inhibit angiogenesis, yet reduced ability to complex with protease inhibitors. Analysis of patient cohorts confirmed prior published findings that rs17632542 SNP is associated with reduced risk of prostate cancer. However, analysis of patients who died in these cohorts revealed a novel and unexpected enrichment of this I163T SNP.

Comments:

I have reviewed several earlier versions of this manuscript. I genuinely thank the authors for their dedication; their manuscript has improved with each version. In the spirit of constructive peer review, I do not propose any new kinds of experiments, and as usual, I did not make any private recommendations to the editors. My evaluation appears consistent with my prior comments. I have 3 requests that I personally would insist on as prerequisites for a strong publication and inclusion in this journal:

- a. I would insist that in Figure 1B, the "LNCaP control" 72-hour measurement be added to the bar graph. (Currently, the data can only be inferred from Supplementary Figure 1C and the reviewer rebuttal letter graph).
- b. Similarly, I would insist on adding an analogous "LNCaP control" migration measurement to Figure 1E.
- c. I applaud the authors for having performed the PSA/KLK3 ELISA. It's important contextual data, and I think Figure 1 can only be correctly interpreted if they graphed the ELISA tabular results, and included it directly in main Figure 1.

This study tests the hypothesis that mono/biallelic germline variations of PSA/KLK3 have alternate endogenous activity compared to the wild-type allele. In the only KLK3-expressing model analyzed (predominantly the in vitro studies of Figure 1), my conclusion is that endogenous KLK3 does not have much/any biological activity, as evidenced by shRNA knockdown. In that setting, subsequent overexpression of 5X levels of wt PSA increases the confluence of cells in vitro from 60% confluence to 70% at 96 hours, whereas 5X expression of PSA I163T decreases it to 55%. Would mono/biallelic (0.5X-1X) expression of PSA I163T have meaningfully distinct biology from wt PSA in an in vivo xenograft assay or in a patient? I don't know, and my understanding is that additional in vivo studies are not a prerequisite to consider revision. By making the above changes to panels 1B, 1E and including the ELISA, the field can analyze the authors' data and draw their own conclusions.

Reviewer #3 (Remarks to the Author):

A PSA SNP associates with cellular function and clinical outcome in men with prostate cancer (Revised).

Srinivasan et al. have described an interesting study investigating a germline I163T variant of PSA and its association with prostate cancer outcomes. In an earlier manuscript version, the authors demonstrated that high overexpression of this PSA variant decreased proliferation in two KLK3-negative lines (PC3, MSK-PCa3) and unexpectedly increased metastatic-associated traits in vitro and in vivo. In subsequent revisions, they have added a KLK3-expressing cell line (LNCaP). They went on to demonstrate in vitro that the I163T variant has reduced peptidase activity, reduced ability to inhibit angiogenesis, yet reduced ability to complex with protease inhibitors. Analysis of patient cohorts confirmed prior published findings that rs17632542 SNP is associated with reduced risk of prostate cancer. However, analysis of patients who died in these cohorts revealed a novel and unexpected enrichment of this I163T SNP.

Comments:

I have reviewed several earlier versions of this manuscript. I genuinely thank the authors for their dedication; their manuscript has improved with each version. In the spirit of constructive peer review, I do not propose any new kinds of experiments, and as usual, I did not make any private recommendations to the editors. My evaluation appears consistent with my prior comments. I have 3 requests that I personally would insist on as prerequisites for a strong publication and inclusion in this journal:

Thank you for acknowledging our efforts addressing Reviewers' comments in our previous versions.

- a. I would insist that in Figure 1B, the "LNCaP control" 72-hour measurement be added to the bar graph. (Currently, the data can only be inferred from Supplementary Figure 1C and the reviewer rebuttal letter graph).

We have added the LNCaP control data to Figure 1B in the revised version.

- b. Similarly, I would insist on adding an analogous "LNCaP control" migration measurement to Figure 1E.

We have not included the LNCaP control, but LNCaP vector control into our migration analysis due to limited availability of inserts for this assay and as we considered shKLK3+vector as our primary control for comparison.

- c. I applaud the authors for having performed the PSA/KLK3 ELISA. It's important contextual data, and I think Figure 1 can only be correctly interpreted if they graphed the ELISA tabular results, and included it directly in main Figure 1.

This table is now included as Figure 1A.

This study tests the hypothesis that mono/biallelic germline variations of PSA/KLK3 have alternate endogenous activity compared to the wild-type allele. In the only KLK3-expressing model analyzed (predominantly the in vitro studies of Figure 1), my conclusion is that endogenous KLK3 does not have much/any biological activity, as evidenced by shRNA knockdown. In that setting, subsequent overexpression of 5X levels of wt PSA increases the confluence of cells in vitro from 60% confluence to 70% at 96 hours, whereas 5X expression of PSA I163T decreases it to 55%. Would mono/biallelic (0.5X-1X) expression of PSA I163T

have meaningfully distinct biology from wt PSA in an in vivo xenograft assay or in a patient? I don't know, and my understanding is that additional in vivo studies are not a prerequisite to consider revision. By making the above changes to panels 1B, 1E and including the ELISA, the field can analyze the authors' data and draw their own conclusions.

We have consistently demonstrated the effect of the SNP on the cellular function of prostate cancer cells using three different cell line models, which express varying levels of endogenous PSA and AR, and we have utilized both overexpression (OE) and knockdown (KD) approaches. Our cell line data is consistent with the findings from recombinant PSA isoforms, which also showed a significant difference in the biochemical activity of PSA and patient sample data. We also acknowledge the limitations of our in vitro and in vivo models employed in the study that may not fully recapitulate the complex tumour microenvironment or the impact of AR signalling or inhibition in cells expressing Thr¹⁶³ PSA. Further investigation addressing these limitations could provide more definitive answers, a point that we have critically discussed in our discussion section (page 12, lines 390-403).